

# Microanatomy and paleohistology of the intercentra of North American metoposaurids from the Upper Triassic of Petrified Forest National Park (Arizona, USA) with implications for the taxonomy and ontogeny of the group

Bryan M. Gee[1], William G. Parker[2] and Adam D. Marsh[2]

[1] Department of Biology, University of Toronto Mississauga, Mississauga, ON, Canada
[2] Division of Science and Resource Management, Petrified Forest National Park, AZ, USA

Corresponding author
Bryan M. Gee,
bryan.gee@mail.utoronto.ca

## ABSTRACT

Metoposaurids are temnospondyl amphibians that are commonly recovered from the Chinle Formation deposits of North America. Two species, *Koskinonodon perfectus* and *Apachesaurus gregorii*, are known from Petrified Forest National Park (PEFO), AZ. Small, elongate intercentra are the single diagnostic postcranial characteristic of the smaller *A. gregorii*. However, a poor understanding of the earliest life stages of *Koskinonodon perfectus* and other large metoposaurids makes it unclear whether the proportions of the intercentra are a diagnostic feature for species discrimination or whether they are influenced by ontogeny. Previous work on metoposaurid intercentra has shown that ontogenetic information can be extrapolated from histological analyses. Here, we perform an analysis of the microanatomy and the histology of metoposaurid intercentra from PEFO to determine their ontogenetic maturity and in turn whether elongate intercentra are a reliable taxonomic feature for distinguishing North American metoposaurids. Our findings suggest that the elongate intercentra are the result of ontogenetic variation within a single large-bodied metoposaurid taxon rather than interspecific variation between two metoposaurids of vastly different adult sizes. These findings have significant implications for the taxonomy of North American metoposaurids and subsequently for interpretations of the Chinle Basin paleoenvironment. Furthermore, this study provides the first histological characterization of North American metoposaurid intercentra, thereby improving the understanding of vertebral ontogeny within Metoposauridae and offering new insights into the ecology of large metoposaurids.

## INTRODUCTION

Metoposaurids are Late Triassic temnospondyl amphibians with a global distribution and are some of the most commonly recovered fossils from freshwater depositional

settings (e.g., *Hunt, 1993*). They are characterized by dorsoventrally flattened skulls with anteriorly positioned orbits, a highly ossified skeleton, and an adult size in excess of two meters in most taxa (*Colbert & Imbrie, 1956*). There are presently three accepted taxa of metoposaurids in North America: two of large size, *Koskinonodon perfectus* and *Metoposaurus bakeri*, and one of small size, *Apachesaurus gregorii* (*Case, 1922*, *1931*; *Branson & Mehl, 1929*; *Hunt, 1993*; *Mueller, 2007*). The taxonomy of metoposaurids, particularly the North American taxa, is in need of revision, hence the indeterminate affinities of '*Metoposaurus*' *bakeri*, but such an analysis is beyond the scope of this paper. *Koskinonodon perfectus* and *A. gregorii* have a wide distribution in the Southwestern United States, including Arizona, New Mexico, and Texas, while *Metoposaurus bakeri* is mainly restricted to Texas, with a single juvenile specimen from Nova Scotia (*Case 1932*, *Hunt & Lucas, 1993*; *Long & Murry, 1995*; *Heckert & Lucas, 2002*; *Parker & Martz, 2011*). It is worth noting that neither *A. gregorii* nor any other diminutive species of metoposaurid is known outside of North America (*Long & Murry, 1995*; *Spielmann & Lucas, 2012*).

Both *Koskinonodon perfectus* and *A. gregorii* are commonly recovered from the Chinle Formation exposures of Petrified Forest National Park (PEFO), AZ, USA. *Koskinonodon perfectus* is common in the lower units within the Chinle Formation (the Blue Mesa Member and lower part of the Sonsela Member) and is rare in the upper units (the upper part of the Sonsela Member and the Petrified Forest Member) (*Hunt & Lucas, 1993*; *Heckert & Lucas, 2002*; *Martz & Parker, 2010*; *Parker & Martz, 2011*). *A. gregorii* demonstrates the opposite pattern of stratigraphic distribution (*Parker & Martz, 2011*). Although *Koskinonodon perfectus* is known from extensive amounts of cranial and postcranial material, including several dozen well-preserved skulls, the vast majority of material ascribed to *A. gregorii* consists of isolated, elongate intercentra that are considered diagnostic of the taxon (*Hunt, 1993*; *Long & Murry, 1995*). The original diagnosis of *A. gregorii* by *Hunt (1993*: 81–85*)* designated a diameter-to-length ratio of less than 0.8 as "elongate." However, the methodology used by *Hunt (1993)* to measure intercentra was poorly described, no data were presented to support the use of 0.8 as a quantitative threshold, and this metric was not included in the amended diagnosis of *Spielmann & Lucas (2012)*. We also note that elongate intercentra are not exclusive to this taxon, as they are also known in the large metoposaurid *Dutuitosaurus ouazzoui* (*Dutuit, 1976*; *Hunt, 1993*). Furthermore, although the diagnosis of *A. gregorii* includes a total of 11 cranial traits, the entire set can only be confirmed in the holotype specimen, and most other referred specimens preserve only a few of the diagnostic features (*Spielmann & Lucas, 2012*). Finally, whereas size has frequently been used as an informal characteristic in identifying specimens (*A. gregorii* being significantly smaller than all other metoposaurid taxa), this is not a reliable metric given the role of ontogeny in changing body size (e.g., *Horner, De Ricqlès & Padian, 1998*; *Steyer et al., 2004*; *Horner & Goodwin, 2009*; *Werning, 2012*; *Konietzko-Meier & Sander, 2013*). This is further confounded by the absence of a preserved complete growth series for North American metoposaurids, particularly with respect to the earliest life stages. Among the dozens of known skulls of *Koskinonodon perfectus*, only two ascribed to juvenile specimens have been described in detail (*Zanno et al., 2002*; *Gee & Parker, in press*).

 

Although *A. gregorii* has been interpreted to be a small yet mature metoposaurid taxon, it remains unclear whether the diagnostic features of the taxon, such as the shallow otic notches and the elongate intercentra, are the product of speciation or if they are merely a misinterpretation of features influenced by ontogeny in large metoposaurids. Such a possibility is rarely considered in determining whether small metoposaurid specimens are skeletally mature individuals of *A. gregorii*, as argued by *Hunt (1993)*, or skeletally immature individuals of a large metoposaurid such as *Koskinonodon perfectus*. Although *Hunt (1993)* claimed that *A. gregorii* represents a small yet mature taxon, many of the arguments rely heavily on external morphology and a scant sample of juvenile metoposaurid specimens, and its ontogenetic maturity has never been confirmed by more informative methods such as histological analyses. Accordingly, the diagnosis of *A. gregorii*, which relies heavily on elongate intercentra given their prevalence in depositional settings, is tentative in the absence of multiple specimens that can confirm more of the informative cranial features or that can clearly associate elongate intercentra with diagnostic cranial material. In this study, we focus on analyzing the histology and microanatomy of the single diagnostic postcranial trait of *A. gregorii*, elongate intercentra. The goal of the analysis is to elucidate additional ontogenetic information beyond that of the external morphology to better determine whether *A. gregorii* represents a small, mature metoposaurid or if it may in fact be an early ontogenetic stage of a large metoposaurid such as *Koskinonodon perfectus*. If *A. gregorii* were indeed a significantly smaller metoposaurid, we would expect to find histological and anatomical markers of relative maturity (e.g., high degree of secondary remodeling, low abundance of calcified cartilage, presence of annuli with lines of arrested growth (LAGs)) in the diagnostic intercentra.

Bone histology is a common method used to study ontogeny in a variety of extinct taxa, often by comparison to extant members of these clades (summarized in *Padian, 2013*). Although the majority of paleohistological inquiries have centered on amniotes, several analyses have focused on temnospondyls. Most of these studies focus on long bones (e.g., *Damiani, 2000*; *Konietzko-Meier & Schmitt, 2013*; *Konietzko-Meier, Shelton & Sander, 2015*; *Sanchez et al., 2010a*; *Sanchez & Schoch, 2013*; *McHugh, 2014*, *2015*), as is common with other vertebrate clades, but other portions of the skeleton, such as osteoderms (e.g., *Witzmann & Soler-Gijón, 2010*), the dermal sculpting of the skull and pectoral elements (e.g., *Witzmann, 2009*), and teeth (e.g., *Warren & Davey, 1992*; *Warren & Turner, 2005*) have also been examined. Many of the early histological analyses of metoposaurids utilized material of the European *Metoposaurus* and the Moroccan *Dutuitosaurus* (e.g., *Dutuit, 1976*; *Gross, 1934*; *de Ricqlès, 1975*, *1978*, *1979*), and more recent studies have continued to utilize these genera for the study of long bones and cranial bones in the Polish taxon *Metoposaurus krasiejowensis* (*Konietzko-Meier & Sander, 2013*; *Konietzko-Meier & Sander, 2013*; *Gruntmejer, Konietzko-Meier & Bodzioch, 2016*) and long bones of *Dutuitosaurus* (*Steyer et al., 2004*; *Sanchez et al., 2010b*). Less attention has been directed toward the North American taxa (e.g., *de Ricqlès, 1981*; *Warren & Davey, 1992*). Furthermore, histological studies of temnospondyl intercentra have been performed only a handful of times

(e.g., *Enlow & Brown, 1956*; *Ray, Mukherjee & Bandyopadhyay, 2009*; *Mukherjee, Ray & Sengupta, 2010*; *Konietzko-Meier, Danto & Gądek, 2014*; *Danto, Witzmann & Fröbisch, 2016*), and the only previous examination of metoposaurid intercentra was conducted on the European taxon *Metoposaurus krasiejowensis* (*Konietzko-Meier, Bodzioch & Sander, 2013*). Therefore, the study of North American metoposaurid intercentra is important for augmenting existing morphological and histological data that can be used in comparisons of metoposaurid paleoecology and ontogeny. Metoposaurid intercentra spanning a wide size range are commonly recovered elements at PEFO, making them more accessible for histology than the relatively rare limb elements. This study seeks to provide an alternative approach to comparisons of external morphology in order to evaluate the potential for metoposaurid intercentra proportions to be influenced by ontogeny rather than speciation.

## MATERIALS AND METHODS

### Selection of specimens

All the sample specimens were collected from the Late Triassic sedimentary rocks of the Chinle Formation at PEFO, AZ, USA (Figs. 1 and 2). Metoposaurids are found throughout three units of the Chinle (the Blue Mesa Member, Sonsela Member, and Petrified Forest Member), but there are disparate relative abundances of large and small metoposaurids throughout the stratigraphic column (*Parker & Martz, 2011*). Eight of the 10 elements were selected with the goal of sampling a pair of intercentra from the same stratigraphic horizon, if not the same locality: one intercentrum of shortened proportions normally referred to *Koskinonodon perfectus* and one intercentrum of elongate proportions comparable to those normally referred to *A. gregorii* (Table 1; Figs. 1 and 2). As noted in the introduction, the term "elongate" is not quantitatively defined in the most recent diagnosis of *A. gregorii* (*Spielmann & Lucas, 2012*); therefore, our selection of specimens typical of *A. gregorii* was based on specimens that have been figured by past authors (e.g., *Hunt, 1993*; *Long & Murry, 1995*; *Spielmann & Lucas, 2012*).

Elements are assigned to the same specimen number on the basis of physical proximity during collection and general taxonomic identity and should not be interpreted to mean that the elements are from the same individual. We have created letter designations for multi-element collection numbers for the purpose of this study (Table 1). Detailed locality information is on file in the archives at PEFO and is available to qualified researchers. PEFO 4826 and PEFO 38726 are from locality PFV 122 in the Blue Mesa Member (Figs. 1–3). PEFO 38645 is from PFV 040 in the Petrified Forest Member (Figs. 1–3). PEFO 36874 and PEFO 16696 (two and three intercentra, respectively) are from a locality (PFV 215) in the Petrified Forest Member (Figs. 1–3). The final two intercentra, belonging to PEFO 35392 (also from PFV 215), were selected because of their association with a skull of a small metoposaurid that was interpreted to be a juvenile *Koskinonodon perfectus* (*Gee & Parker, in press*; Figs. 1–3). PEFO 4826, PEFO 36874a, and PEFO 16696a are of a typical size and proportion for *A. gregorii* (Fig. 3). PEFO 38726 is typical for *Koskinonodon perfectus* (Fig. 3). Several other elements, although relatively small compared to the largest known intercentra of the taxon, are of proportions closer to

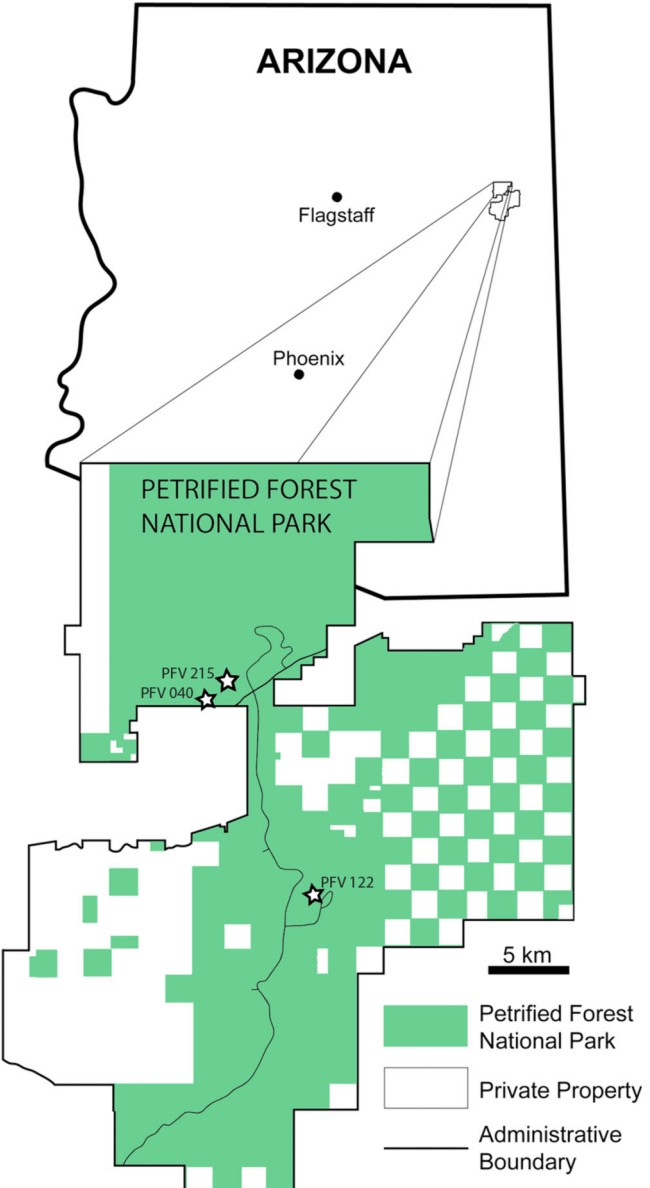

**Figure 1 Map of PEFO showing localities of sampled specimens.** Localities and associated specimens are as follows: PFV 122 (Blue Mesa Member): PEFO 4826 and PEFO 38726; PFV 040 (Petrified Forest Member): PEFO 38645; PFV 215 (Petrified Forest Member): PEFO 36874, PEFO 16696, and PEFO 35392. Modified from *Parker & Irmis (2005)*.

those of *Koskinonodon perfectus*; these include: PEFO 38645, PEFO 36874b, PEFO 16696c, and PEFO 35392 (Fig. 3). PEFO 16696b is the most taxonomically ambiguous, as it is nearly identical in size to PEFO 4826 but is also of different proportions and morphology (Fig. 3). Specimens were measured using the same standards as *Konietzko-Meier, Bodzioch & Sander (2013)* (Fig. 4A). The overall size range of the elements sampled in this study (mediolateral width between 9.81 mm and 55.32 mm) is comparable to that sampled by *Konietzko-Meier, Bodzioch & Sander (2013)*, which ranged from 20.1 to 71 mm (Table 1).

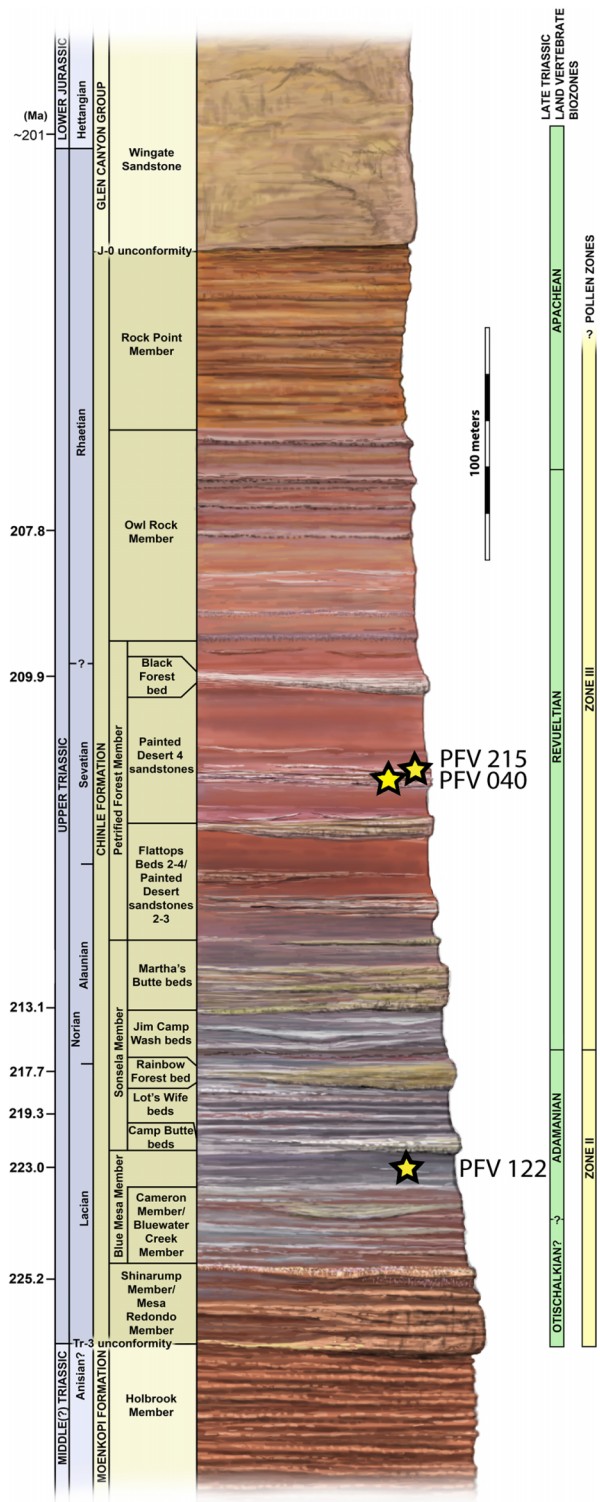

**Figure 2 Stratigraphic column of PEFO showing position of sampled specimens and localities.** Localities and associated specimens are as follows: PFV 122 (Blue Mesa Member): PEFO 4826 and PEFO 38726; PFV 040 (Petrified Forest Member): PEFO 38645; PFV 215 (Petrified Forest Member): PEFO 36874, PEFO 16696, and PEFO 35392. Dates from *Ramezani et al. (2011)*. Biostratigraphic data from *Parker & Martz (2011)* and *Reichgelt et al. (2013)*.

**Table 1 Summary of intercentra analyzed in this experiment.**

| Specimen number | Estimated position | Cutting plane | Length (mm) | Width (mm) | Height (mm) | W:L | Geologic member |
|---|---|---|---|---|---|---|---|
| PEFO 38726 | Anterior dorsal | Sagittal | 22.98 | 55.32 | 46.91 | 2.40 | BMM |
| PEFO 4826 | Dorsal | Sagittal | 10.25 | 10.55 | 12.71 | 1.03 | BMM |
| PEFO 38645 | Presacral | Sagittal | 10.99 | 21.90 | 19.32 | 1.99 | PFM |
| PEFO 36874a | Dorsal | Sagittal | 7.65 | 10.72 | 8.85 | 1.40 | PFM |
| PEFO 36874b | Perisacral | Sagittal | 11.85 | 19.63 | 17.25 | 1.65 | PFM |
| PEFO 35392a | Mid-dorsal | Sagittal | 15.43 | 28.27 | 25.74 | 1.83 | PFM |
| PEFO 35392b | Anterior dorsal | Sagittal | 15.37 | 25.89 | 24.72 | 1.68 | PFM |
| PEFO 16696a | Presacral | Sagittal | 8.22 | 10.22 | 9.09 | 1.24 | PFM |
| PEFO 16696b | Mid-dorsal | Sagittal | 9.52 | 15.96 | 12.11 | 1.67 | PFM |
| PEFO 16696c | Mid-dorsal | Sagittal | 16.60 | 26.83 | 16.13 | 1.61 | PFM |

Notes:

For specimens with multiple elements, the listed order reflects their order by size, from smallest to largest. Letter assignments for multi-element specimens were created for the purpose of this publication to facilitate their references throughout the text. Measurements were performed in the same manner as in *Konietzko-Meier, Bodzioch & Sander (2013)*, where length is in the anteroposterior axis, width is in the mediolateral axis, and height is in the dorsoventral axis.

Geologic member abbreviations: BMM, Blue Mesa Member; PFM, Petrified Forest Member.

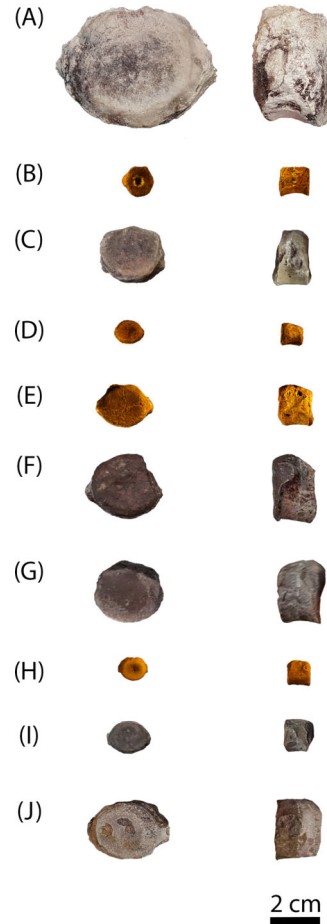

**Figure 3 Photographs of sampled specimens in anterior and lateral profiles.** (A) PEFO 38726; (B) PEFO 4826; (C) PEFO 38645; (D and E) PEFO 36874; (F and G) PEFO 35392; (H–J) PEFO 16696. Order of photographed specimens mirrors their listed order in Table 1. Scale bar = 2 cm.

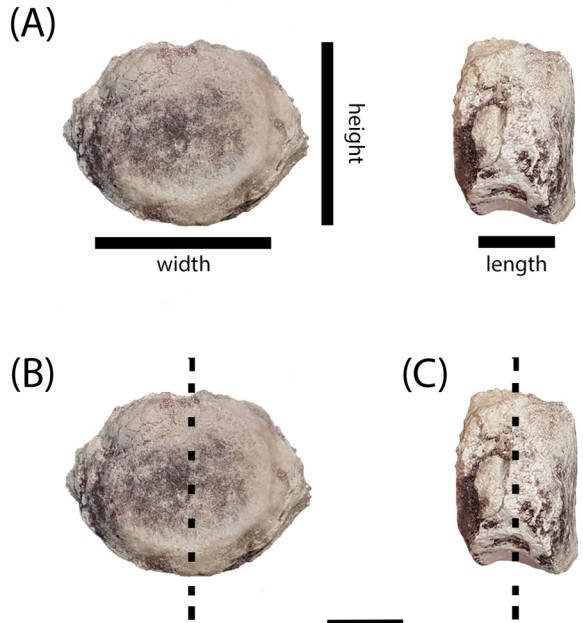

**Figure 4 Measurement positions and sectioning planes of the intercentra.** (A) Measurement positions for length, width, and height of intercentrum (PEFO 38726); (B) sagittal plane of sectioning; (C) transverse plane of sectioning. Scale bar = 2 cm.

## Classification of specimens' axial position

Because North American metoposaurids, especially those from PEFO, are rarely articulated, determining the exact serial position of the studied vertebrae remains difficult. Vertebrae are placed using previously-outlined criteria (*Sulej, 2007*), but it should be noted that these criteria were used in the description of *Metoposaurus krasiejowensis* and it remains unknown what differences may exist in the vertebral column between the European and North American taxa, especially in the absence of preserved neural or hemal arches. Additionally, the determination of axial position in *Metoposaurus krasiejowensis* was made by comparison to the Moroccan metoposaurid *D. ouazzoui*, in which elongate intercentra more comparable to *A. gregorii* than to other metoposaurids are known (*Dutuit, 1976*; *Sulej, 2007*). Finally, intraspecific variation in North American metoposaurids is poorly known; thus the serial position of smaller intercentra is the most tentative.

## Thin section preparation and imaging

The intercentra were first cleaned using a toothbrush and water to remove excess matrix before being consolidated with Paraloid B-72 (Rohm and Haas, Philadelphia, PA, USA) dissolved in acetone. All specimens were molded and casted according to National Park Service museum standards (*National Park Service, 2007*), with Carbowax (molecular weight 4,000; Dow Chemical, Midland, MI, USA) added to stabilize cracks and other fragile areas. After creating two-part molds using Platsil 73-25 silicone rubber, the Carbowax was removed using a brush and warm water. All specimens were impregnated in a polyester resin mixture of Castolite™ AC and hardener (Eager Polymers, Chicago, IL, USA)

at a ratio of 1 oz of Castolite$^{TM}$ to 12 drops of hardener. The specimens were placed in a vacuum chamber to evacuate gas from the resin and then allowed to cure for a minimum of 24 h. Because the primary focus of the study was to assess the ontogenetic stage of various intercentra to determine whether small, elongate intercentra ascribed to *A. gregorii* belonged to juveniles of *Koskinonodon perfectus*, we initially focused on sagittal cuts (down the midline in the anteroposterior axis) (Fig. 4B), but we subsequently made transverse (mediolateral axis) sections of all specimens (Fig. 4C). All specimens were cut using an automated IsoMet 1000 Precision Saw (Buehler, Bluff, IL, USA). The cut surface of the desired block and its respective thin section were prepared by polishing each with a 600-mesh silicon carbide on glass plates. Both surfaces were rinsed with ethanol and then attached to plexiglass slides using Scotch–Weld Instant Adhesive (CA40; 3M). The sections were allowed to dry for a minimum of 1 h. All specimens except PEFO 38726 were cut to a height of 0.7 mm using the IsoMet 1000 Precision Saw. PEFO 38726 was too large to be cut by the automatic saw, so it was cut manually with a larger saw fitting for the IsoMet. All specimens were polished in the following sequence: Hillquist 1010 grinding cup, 600-mesh grit, 1,000-mesh grit, and one micron grit. PEFO 38726 was polished on a 600-mesh lap wheel before polishing on the Hillquist to remove uneven surfaces from the manual cut. The thin sections were gradually ground down with repeated examination under a compound microscope to evaluate their optical clarity. All polishing after the Hillquist step was done manually on glass plates. Thin sections were imaged on a Nikon Instruments AZ100 Multizoom microscope fitted with AZ-Plan Apo 0.5× and AZ-Fluor 5× objective lenses, an AZ-RP rotatable polarizer plate, and a DS-Fi2 digital camera mount. NIS-Elements imaging software was used for this study.

## RESULTS

### General microanatomy and histology

Overall, the microanatomy and histology of the intercentra sampled are very similar to those that were described for *Metoposaurus krasiejowensis* (*Konietzko-Meier, Bodzioch & Sander, 2013*) (Figs. 5–7).

The microanatomy of the sampled intercentra is consistent within the sample regardless of size or any variation in axial position. Nearly the entire element is made of endochondral and periosteal trabecular bone; lamellar bone is found only in thin layers deposited within the vascular canals and in the compact annulus at the ventral margin in mature individuals. In sagittal profile, most of this tissue is endochondral bone that is found at the dorsal, anterior, and posterior margins in all specimens and at the ventral margin in more ontogenetically immature individuals (Figs. 5A and 5C). A periosteal cortex is located in the ventral region of the element with the corners near the anteroventral and posteroventral corners and the apex terminating near the geometrical center (Figs. 5A and 5C). In transverse profile, the endochondral bone is found in the dorsal and more interior regions of the element (Figs. 5B and 5D). The periosteal cortex is found at the ventral margin and extends up the lateral margins in this profile (Figs. 5B and 5D). *Konietzko-Meier, Bodzioch & Sander (2013)* noted the presence of periosteal bone in the dorsal half of a presacral intercentrum and the atlas in

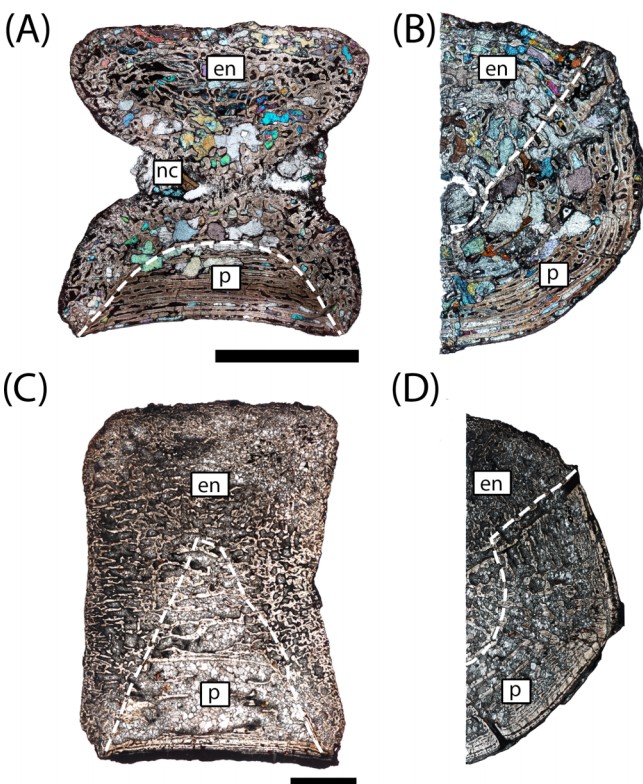

**Figure 5 Microanatomy of sectioned intercentra.** (A) A typical small intercentrum, PEFO 16696a, in sagittal profile; (B) the same specimen in transverse profile; (C) a typical large intercentrum, PEFO 16696c in sagittal profile; (D) the same specimen in transverse profile. en, endochondral bone; nc, notochordal canal; p, periosteal bone. Scale bars = 4 mm.

transverse section, but an atlas was not sampled in this study, and no periosteal bone was definitively noted in the dorsomedial region of the presacral intercentra (PEFO 16696a, PEFO 38645). Within the periosteal region, the trabeculae are distributed in an orderly set of layers that are parallel to the ventral margin of the intercentrum, usually giving them a slightly convexity (Figs. 5 and 6). In the smallest intercentra, the periosteal region is semi-circular in sagittal profile owing to the presence of an open notochordal canal (Figs. 5A, 8 and 9). In larger intercentra, the cortex is triangular with an apex that terminates near the geometric center (Fig. 5C), with the highest apex found in the largest specimen. In sagittal view, the periosteal region is divided from the remainder of the intercentrum by a prominent line of trabeculae that are oriented at a consistent oblique angle that contrasts with the otherwise irregularly distributed endochondral trabeculae (Figs. 5A and 5C). The division of periosteal and endochondral bone is less defined near the apex. Many of the specimens sampled by this study have been partially infilled, and in some cases slightly damaged, by secondary precipitation of carbonate minerals. In most specimens, the minerals appear to have precipitated within the existing internal structures, although some parallel layers of the periosteal cortex have been lost in larger specimens.

The histological nature of the metoposaurid intercentrum is more variable within the sample, differing primarily in the relative abundance and distribution of various

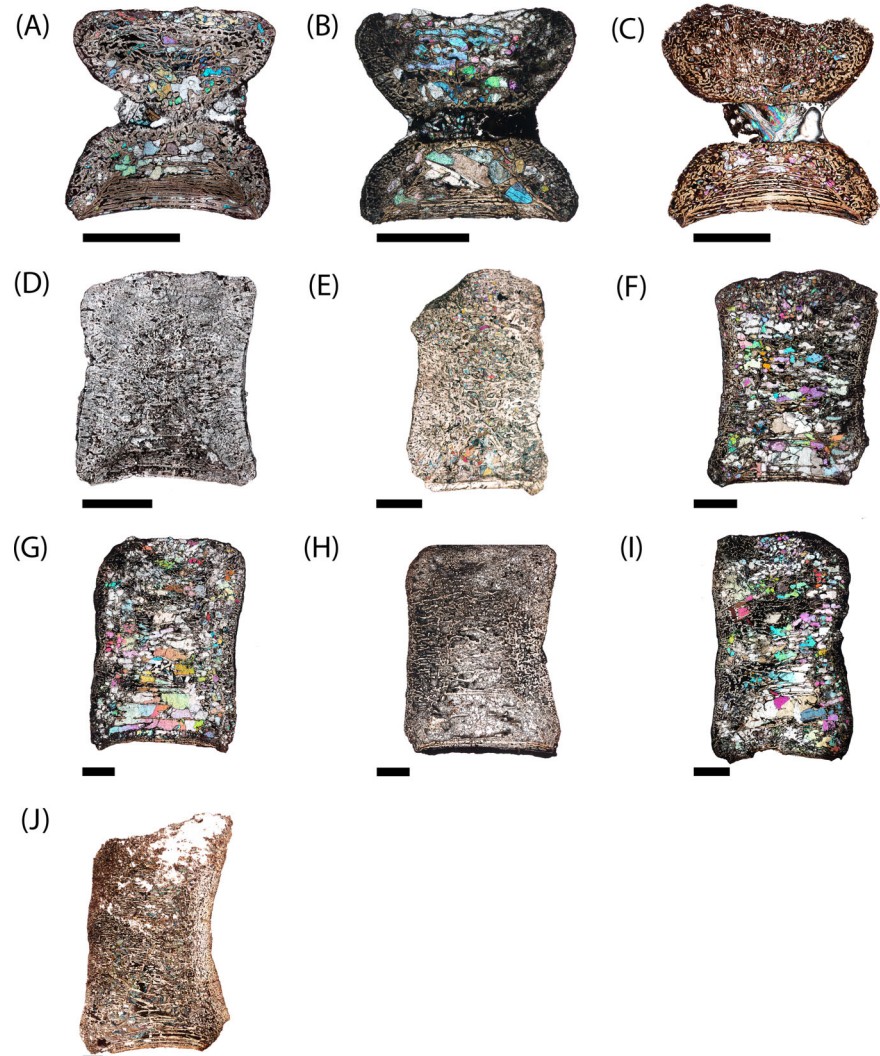

**Figure 6 Microphotographs of the sagittal sections of sampled specimens.** (A) PEFO 36874a; (B) PEFO 16696a; (C) PEFO 4826; (D) PEFO 16696b; (E) PEFO 38645; (F) PEFO 36874b; (G) PEFO 35392a; (H) PEFO 16696c; (I) PEFO 35392; (J) PEFO 38726. Scale bars = 4 mm.

features such as calcified cartilage and the deposition of primary lamellar bone rather than the presence or absence of these features. Although the majority of each intercentrum is made of trabecular bone, more compact lamellar bone can be found deposited inside of vascular canals and in the outer layers of the periosteal cortex. The trabeculae form a denser network with smaller intertrabecular spaces at the articular faces whereas the spacing in the interior and dorsal regions is greater (Fig. 10). The most ventral portions of the periosteal cortex are parallel-layered in both sagittal and transverse profiles, a structure that results from the large numbers of vascular canals oriented in the anteroposterior axis. The external lamellar portion of the cortex (annulus) is poorly vascularized and more compact. LAGs can be identified in this layer in the largest specimen (Fig. 11G). The external periosteal trabeculae form a layered matrix that is oriented parallel to the external margins, but the interior portions of the cortex consist of large erosion cavities with

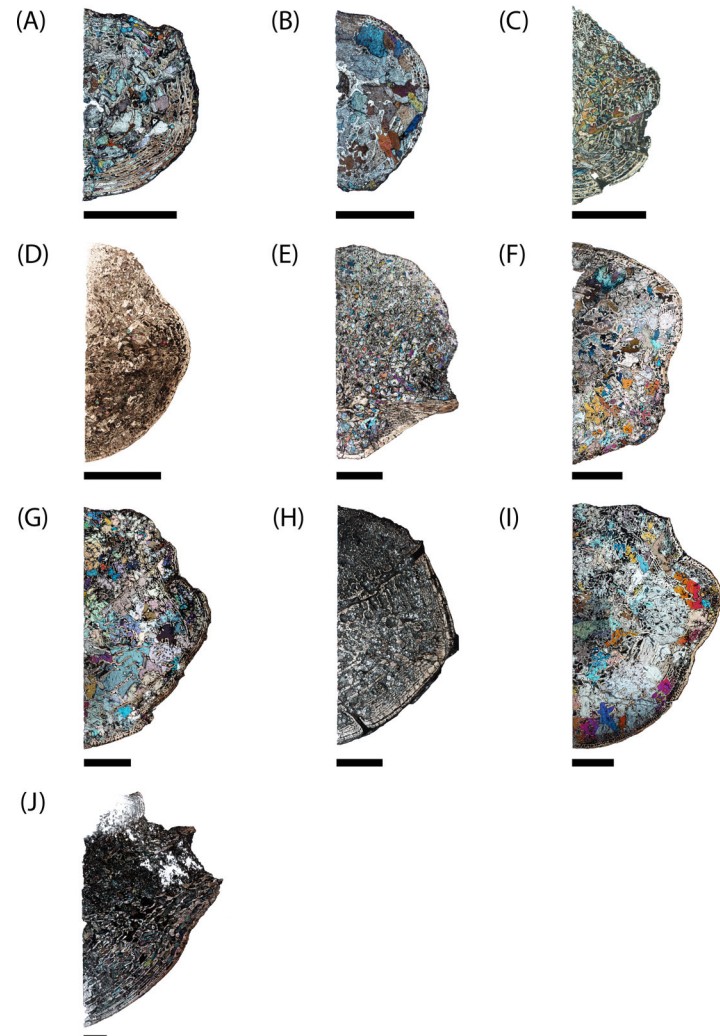

**Figure 7 Microphotographs of the transverse sections of sampled specimens.** (A) PEFO 36874a; (B) PEFO 16696a; (C) PEFO 4826; (D) PEFO 16696b; (E) PEFO 38645; (F) PEFO 36874b; (G) PEFO 35392a; (H) PEFO 16696c; (I) PEFO 35392; (J) PEFO 38726. Note that because all specimens were first sectioned in the sagittal plane, all transverse sections are only half-sections. Scale bars = 4 mm.

irregular networks of trabeculae. These periosteal trabeculae transition to endochondral trabeculae in the innermost portions of the intercentrum. In relatively immature individuals, the trabeculae are frequently associated with calcified cartilage at both the margins and in the interior of the element. In more mature individuals, the cartilage decreases in abundance in conjunction with an increase in secondary bone deposition, particularly within the interior endochondral region. Calcified cartilage persists throughout the entire sample, but becomes rare in the largest intercentrum (Fig. 12). Osteocyte lacunae are primarily circular except in the compact layers of the periosteal cortex in which they are flattened and more elongate. Most intercentra feature large open cavities of various degrees of penetration into the interior of the element that are distributed randomly along the external periosteal margin. These are probably nutrient foramina in which part of the canal is exposed two-dimensionally, accounting for the

variable depth and morphology across the sample. Sharpey's fibers can be found at the ventrolateral margins.

## Detailed descriptions

The following descriptions are listed in order by size, as in Figs. 6 and 7. Because the primary goal of this study was to assess the ontogenetic maturity of small, elongate intercentra normally assigned to *A. gregorii*, the description focuses mainly on features that have been recognized as ontogenetically informative in *Metoposaurus krasiejowensis* (*Konietzko-Meier, Bodzioch & Sander, 2013*).

PEFO 36874a (Figs. 6A and 7A): In sagittal profile, both anterior and posterior articular faces are markedly amphicoelous. The notochordal canal, which extends longitudinally in the anteroposterior axis, is partially infilled with secondary minerals but was also in the process of closure at the time of death via deposition of cartilage into the canal from the dorsal and ventral halves (Figs. 8C and 8D). Owing to the presence of the notochordal canal, the periosteal cortex is semi-circular in shape and originates at the ventral corners (Fig. 9A). The cortex features a large number of vascular canals that are filled with thin primary depositions of lamellar bone. In comparison to larger, presumably more mature intercentra, the total thickness of this lamellar bone is relatively thin. There is no evidence of a compact lamellar annulus in the outer margin of the cortex. The apex of the periosteal cortex is poorly defined in sagittal profile. The cortex is overall highly vascularized. A large nutrient foramen is located slightly ventral to the periosteal-endochondral margin; a possible second, similarly sized foramen is located between these two landmarks. A significantly smaller foramen is found on the ventrolateral surface. The closure of the notochordal canal is a feature not seen in any of the other sampled specimens. The majority of the deposition into the canal is cartilaginous, although some trabeculae can be seen forming closer to the distinct margins that bound the canal. The deposition appears to be originating close to the geometric center and from both sides, although the relative contribution from the dorsal half appears to be greater.

The endochondral trabeculae are thickest at the articular surfaces and the ventral half of the element. They are thinner and more widely spaced in the dorsal endochondral region and toward the geometric center (Fig. 10A). The notochordal canal is bordered on the margins by a dorsoventrally short layer of thick endochondral trabeculae that separates it from the periosteal cortex in the ventral half and from the thinner endochondral network in the dorsal half; the thickness of this layer is approximately the same on both margins and slightly thinner than that at the articular surfaces. Secondary remodeling can be found in the trabeculae at the margin between the periosteal cortex and the endochondral trabecular network, particularly at the dorsal margin of the cortex where this margin becomes less clearly defined, but secondary osteons are extremely rare and found only in the interior of the element and at the precise margins. The vast majority of the trabecular network is primary bone. Calcified cartilage is pervasive at both the margins and the interior of the element.

PEFO 16696a (Figs. 6B and 7B): The microanatomy and histology of PEFO 16696a are similar to those of PEFO 36874a. The most significant differences pertain to the

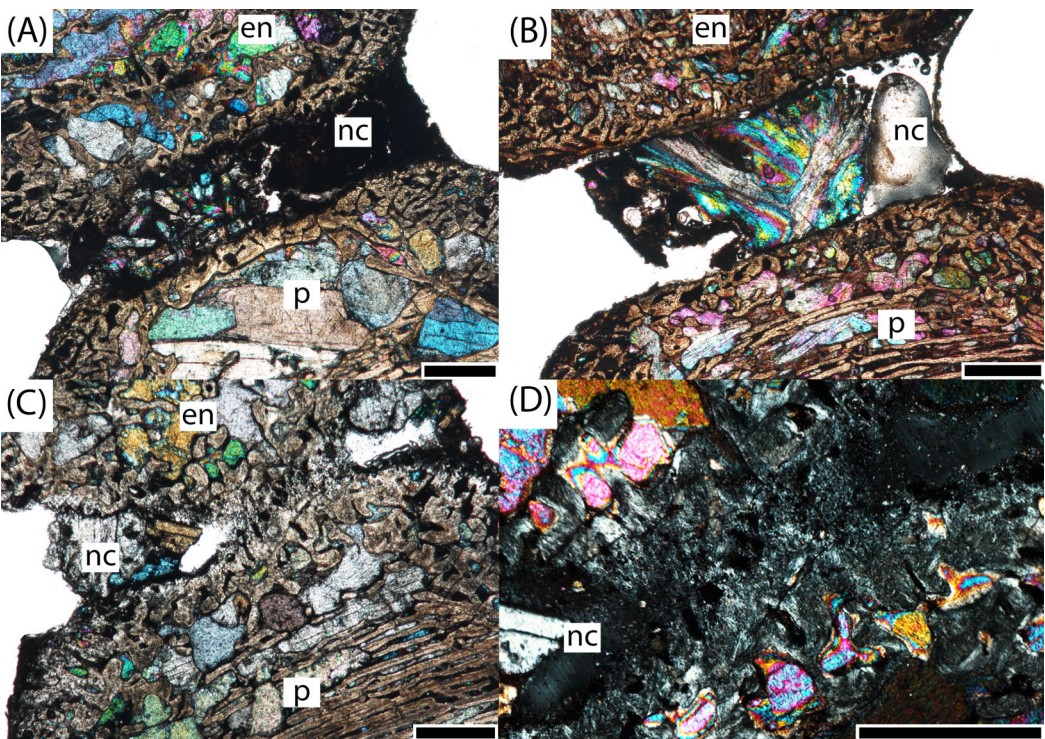

**Figure 8 Microphotographs of the notochordal channel in three small specimens in sagittal profile.**
(A) PEFO 16696a; (B) PEFO 4826; (C) PEFO 36874a; (D) close-up of the closure of the notochordal canal in polarized light. en, endochondral bone; nc, notochordal canal; p, periosteal bone. Scale bars = 1 mm.

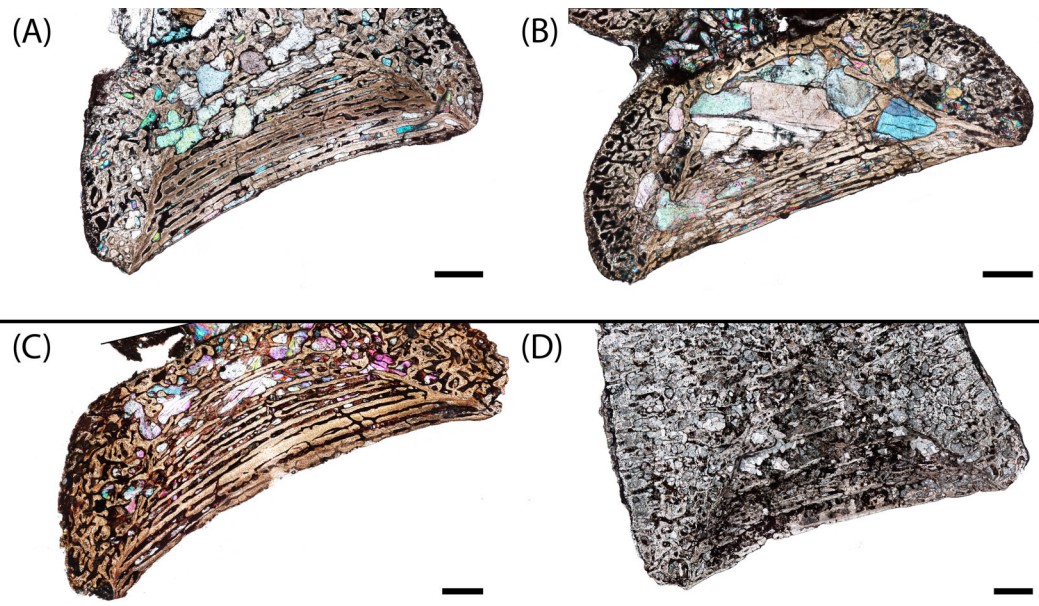

**Figure 9 Microphotographs of the periosteal cortex in four small specimens in sagittal profile.**
(A) PEFO 36874a; (B) PEFO 16696a; (C) PEFO 4826; (D) PEFO 16696b. Scale bars = 1 mm.

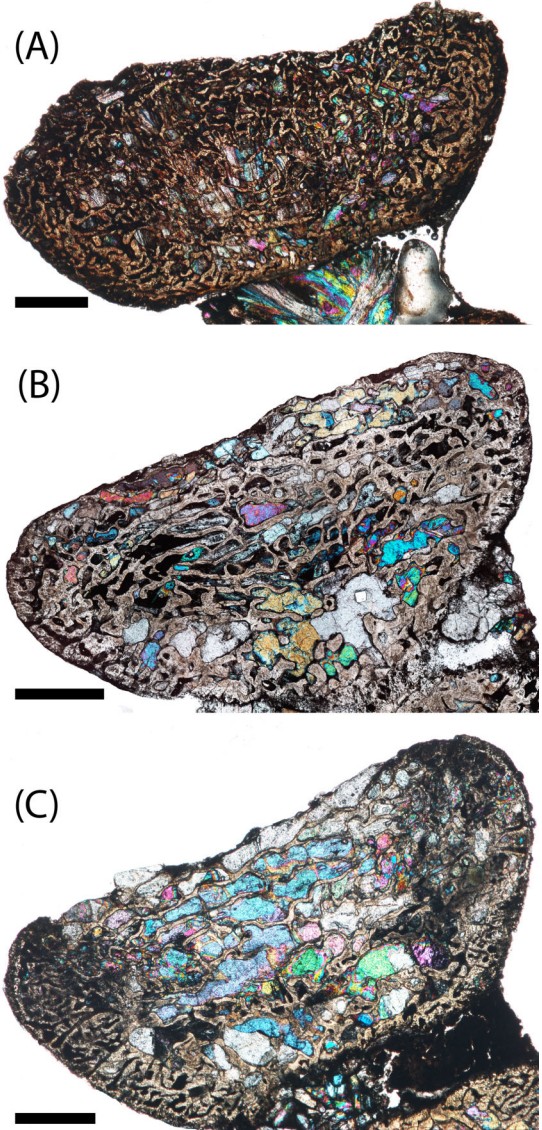

**Figure 10 Microphotographs of the dorsal endochondral region in three small specimens in sagittal profile.** (A) PEFO 4826; (B) PEFO 36874a; (C) PEFO 16696a. Scale bars = 1 mm.

density of the dorsal endochondral network, the vascularization of the periosteal cortex, and the overall quality of preservation. As with the other small intercentra, the total deposition of lamellar bone within vascular canals is relatively thin, and there is no compact lamellar annulus. The only difference in the cortex from that of the other two specimens is a greater spacing between the parallel layers when viewed in transverse profile; no such difference is apparent in sagittal profile. There is no evidence of closure of the notochordal canal as in PEFO 36874a (Fig. 8A). As with the other two specimens, the notochordal canal is bounded on both margins by a short layer of thick endochondral trabeculae.

The spacing between the endochondral trabeculae is the greatest of the sampled specimens and does not appear to be a feature of damage from diagenesis, at least in

sagittal profile (Fig. 10B). In this view, only the dorsal portion of the periosteal cortex is noticeably more damaged than in PEFO 36874a where some of the parallel layers have been completely destroyed by coarse-grained secondary carbonates. However, in transverse profile, nearly all of the internal bone has been destroyed by similarly coarse carbonates, preserving only a sparse scattering of endochondral trabeculae in the dorsal portion of the intercentrum, the trabeculae immediately surrounding the notochordal canal, and external most layers of the periosteal cortex. Enough of the obliquely oriented trabeculae are preserved to identify the distinct margin between endochondral and periosteal domains. A small degree of secondary remodeling can be identified, particularly at this margin and around the notochordal canal, but its relative abundance compared to primary deposition cannot be resolved in transverse profile. In sagittal profile, secondary remodeling and secondary osteons are relatively rare. At least one nutrient foramen can be identified in transverse profile in the cortex, but the extent of penetration is indeterminate. Where trabeculae are preserved, calcified cartilage is pervasive, including at both the center near the notochordal canal and at the outer margins.

PEFO 4826 (Figs. 6C and 7C): The microanatomy and histology of PEFO 4826 are nearly identical to that of PEFO 36874a and PEFO 16696a, with the only major differences pertaining to the overall degree of vascularization in the periosteal cortex, the density of the endochondral trabecular network, and the partial closure of the notochordal canal. The periosteal cortex is semi-circular, originating at the ventral corners, is comprised of highly vascularized parallel layers with thin depositions of lamellar bone within the vascular canals, grades into endochondral trabeculae dorsally, and is lacking in a compact lamellar annulus. The apex is more clearly defined than in the other two specimens (Fig. 9C). The notochordal canal is infilled with secondary minerals and shows no evidence of closure (Fig. 8B). Its margins are nearly smooth and there is no evidence of tissue deposition into the canal. In transverse profile, two to three open cavities are present in the periosteal cortex; they are relatively shallow, terminating around the transition to large erosion cavities in the interior of the intercentrum. Two of these cavities also appear to extend a short distance horizontally between the parallel layers of the periosteal matrix after penetrating to the same depth within the element. Sharpey's fibers can be identified on the ventrolateral surface; they may be more widely spread, but the external most layer of the cortex appears to have been damaged in many areas.

The endochondral region of this specimen is nearly identical to that of the other two small intercentra in being thickest at the articular surfaces and thinnest in the dorsal region. It differs only in having a slightly less disperse trabecular network in the dorsal portion of the element (Fig. 10C). As with those intercentra, secondary remodeling and secondary osteons are very rare, and calcified cartilage is widespread throughout the element, though to a lesser degree than in the other two specimens.

PEFO 16696b (Figs. 6D and 7D): This specimen is the smallest intercentrum that shows no clear evidence of a notochordal canal. As a result of the absence of the canal, the periosteal cortex is triangular in sagittal profile, with the margins defined by remodeled, obliquely oriented trabeculae and an apex terminating around the mid-height of the

element (Fig. 9D). The cortex is slightly less vascularized than any of the intercentra with notochordal canals. It features a thin deposition of lamellar bone within the vascular canals that is not appreciably different from the smaller intercentra, and there is no evidence of a compact lamellar annulus. A few shallow indentations on the periosteal margin may be nutrient foramina, but there are no large cavities as in some of the other intercentra. A very shallow indentation on the anterior articular surface at about the mid-height of the element (aside from the general concavity of the surface) may be the vestiges of the notochordal canal.

The periosteal trabeculae gradually transition from an orderly parallel-layered matrix to more irregularly spaced trabeculae separated by large erosion cavities before reaching the interior endochondral trabeculae. The endochondral trabeculae are similar to the smaller specimens, being thickened at the margins and thinnest at the dorsal region in sagittal profile, although the intertrabecular spacing is slightly larger, particularly at the dorsal region, and the trabeculae at the articular surfaces are noticeably thicker than in the smaller intercentra. Secondary remodeling is more pronounced, although it still represents a small minority in comparison to primary trabeculae. Secondary osteons, while also relatively rare, are noticeably more abundant than in the smaller intercentra. Calcified cartilage is still abundant throughout the element, and the relative abundance in this specimen to that of the smaller intercentra is not appreciably different.

PEFO 38645 (Figs. 6E and 7E): This specimen is unique in possessing an unusual combination of features not seen in any other specimen. The periosteal cortex is generally similar to that of other large intercentra in being triangular in sagittal profile, but the apex, which is easily identified, terminates well below the mid-height of the element. Most of the cortex has been destroyed except for in the margins, which include a compact lamellar annulus (Fig. 11A). This is the first specimen to possess such an annulus, although there are no LAGs present (Fig. 11A). In transverse profile, the cortex is mainly confined to the ventral surface (Fig. 7E). It appears that the lateral and dorsolateral margins have been slightly weathered during preservation, but the lateral extent of endochondral bone is still far greater than expected. However, one layer of organized vascular canals in periosteal trabeculae can be identified on the dorsal margin, suggesting that the cortex does extend onto this margin as noted in a presacral intercentrum analyzed by *Konietzko-Meier, Bodzioch & Sander (2013)*. We consider this observation more tentative given the lower preservational quality of the specimen and the margins in particular. Sharpey's fibers can be identified in the ventrolateral portion of the cortex.

In addition to being more laterally extensive than in other intercentra, the endochondral trabeculae are extremely thick and densely packed at the anterior surface, both in comparison to all other intercentra of this sample and in comparison to the posterior surface in which the organization is more comparable to other sampled specimens. Although the trabeculae of the dorsal region are thinner, as in other specimens, the absence of the lateral periosteal cortex results in this network extending to the dorsolateral surface of the element. Of particular interest is an unusual clustering of

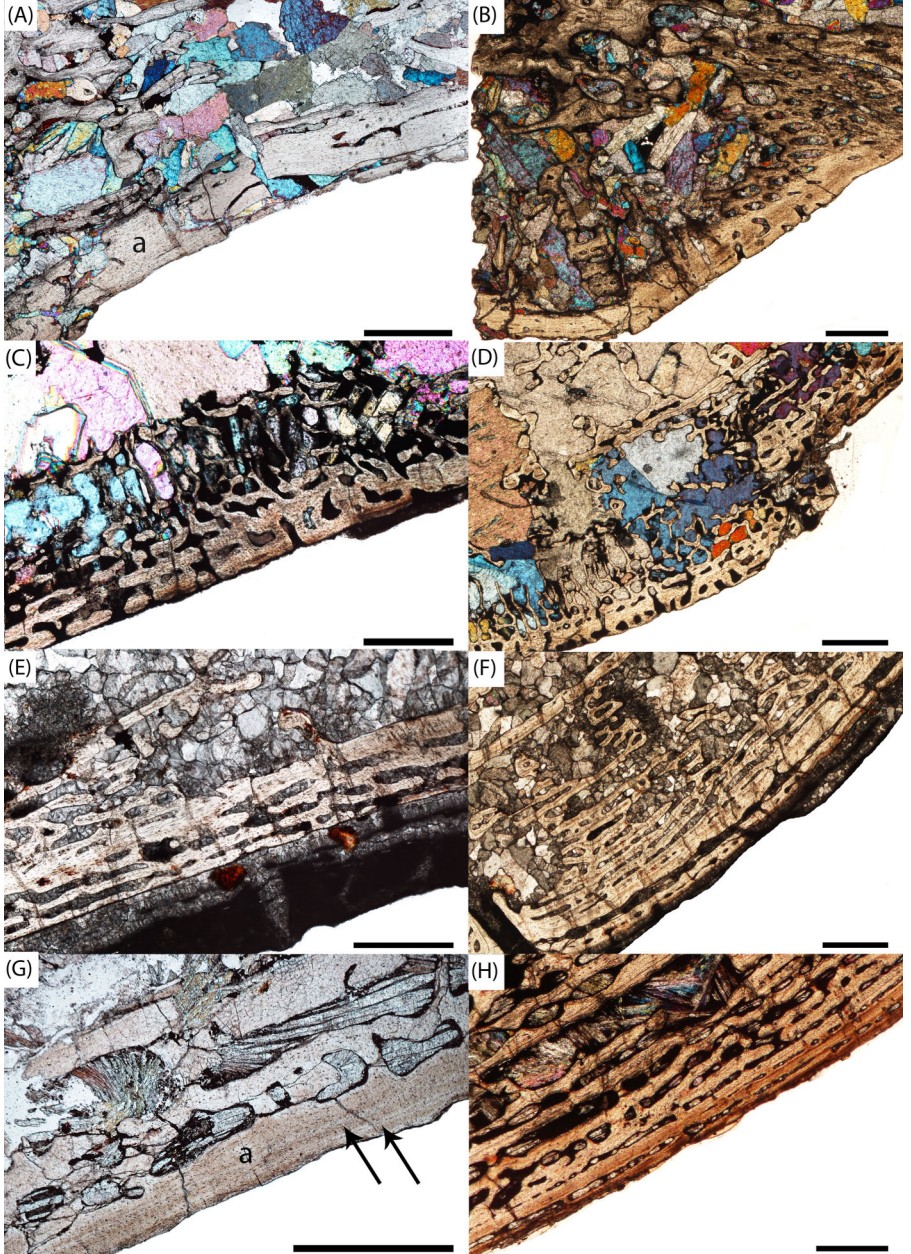

**Figure 11 Microphotographs of the external cortex in large intercentra.** (A) PEFO 38645 in sagittal profile; (B) the same specimen in transverse profile; (C) PEFO 35392a; (D) the same specimen in transverse profile; (E) PEFO 16696c; (F) the same specimen in transverse profile; (G) PEFO 38726; (H) the same specimen in transverse profile. Arrows indicate the position of the LAGs in PEFO 38726. a, annulus. Scale bars = 1 mm.

trabeculae in the dorsal portion of the periosteal cortex, closer to the interior of the element in which a high degree of remodeling is noted. This is connected to a prominent mediolateral line of thickened trabeculae that extends to the external margin. Overall, secondary remodeling is observed at a higher rate than in smaller intercentra, but there is still a significant amount of calcified cartilage within the interior and margins of the element.

PEFO 36874b (Figs. 6F and 7F): This specimen is more similar to the smaller PEFO 16696b than to PEFO 38645. The periosteal region is triangular in sagittal profile with a well-defined apex that terminates below the mid-height of the element, though not to the same degree as PEFO 38645. As with PEFO 16696b, the periosteal cortex maintains a constant thickness up the lateral margin. Most of the parallel layers have been destroyed, and there is no evidence of a compact lamellar annulus, as in PEFO 38645, nor does this appear to be a preservational artifact. The vascularization of the external layers is still relatively high and the deposition of lamellar bone within the vascular canals is only slightly thicker than in PEFO 16696b. Several smaller nutrient foramina are found throughout the periosteal cortex.

The endochondral trabecular network is essentially identical to that of PEFO 16696b. They are thickest at the anterior and posterior surfaces, though to a significantly lesser degree than PEFO 38645, and thinnest in the dorsal region of the element. Calcified cartilage is found in both the interior and at the margins, but it is more sporadic in the interior than in any of the smaller intercentra, and slightly more secondary remodeling can be observed.

PEFO 35392a (Figs. 6G and 7G): This element is similar to PEFO 36874a. The periosteal cortex is nearly identical in shape and vascularity, but it is difficult to locate the apex in this specimen. The lowest possible apex is located slightly ventral to the mid-height of the element where a thickened fragment of trabeculae is located, but the precipitation of large secondary carbonates that are found in the cortex but not the endochondral domains of many other large intercentra continues dorsal to this. There is no evidence of an unusually thick or robust periosteal region in transverse profile. There is also no evidence of a compact annulus, similar to PEFO 36874b (Fig. 11B). The primary lamellar layers found within the vascular canals are noticeably thicker than in the smallest intercentra. Two nutrient foramina are located on the dorsolateral margin of the periosteal cortex. The endochondral region is also similar to the aforementioned intercentra. Calcified cartilage is still present in fairly high abundance, but the degree of both cartilage and secondary remodeling is markedly different than in the smallest intercentra.

PEFO 16696c (Figs. 6H and 7H): This element is similar to all of the large intercentra that lack an open notochordal canal except PEFO 38645 in terms of the organization of the periosteal and endochondral bone. The apex of the periosteal cortex can be more readily identified than in PEFO 35392b and terminates approximately at the mid-height of the element. The cortex is still relatively vascularized, though to a lesser degree than in PEFO 35392b, and there is no evidence of a compact lamellar annulus (Fig. 11C). Two prominent nutrient foramina are present; these are more deeply penetrating than seen in any other intercentra. Calcified cartilage occurs sporadically throughout the interior and remains abundant at the margins. Secondary remodeling is comparable to that of PEFO 35392b.

PEFO 35392b (Figs. 6I and 7I): This specimen is nearly identical to PEFO 35392b and PEFO 16696c in structure and organization. The apex of the periosteal cortex terminates at the mid-height of the element. The external cortex is more damaged in sagittal
profile than in PEFO 35392ba although the external most layer, consisting of a highly vascularized network, is well preserved in transverse profile and is nearly identical to that of PEFO 35392a. Conversely, the interior endochondral region is slightly better preserved in this specimen. There is no evidence of a compact lamellar annulus, even in the transverse profile. One nutrient foramen can be identified on the dorsolateral surface. The relative abundance of calcified cartilage and secondary remodeling is comparable to that of the other two specimens.

PEFO 38726 (Figs. 6J and 7J): This specimen is similar to all other large intercentra regarding the microanatomy; the periosteal cortex is triangular in sagittal profile, terminates slightly above the mid-height of the element, and extends up the lateral margins in transverse profile. The preservational condition of the element is slightly worse than in the other intercentra due to the presence of fibrous bundles of secondary minerals that have damaged the interior trabecular networks.

The external layers of the cortex are less vascularized, particularly on the lateral margins, and the endochondral trabeculae are thicker at the articular surfaces in sagittal profile, particularly on the anterior surface. This is somewhat comparable to the condition of PEFO 38465, but the degree of thickening relative to the overall length of the element is less in this specimen. A compact annulus is present in which two continuous lines of a distinctly different coloration to the periosteal bone that are oriented parallel to the ventral margin are visible in sagittal profile (Fig. 11D). There is no evidence of any large nutrient foramina in the external periosteal margin. Calcified cartilage is extremely rare in comparison to the rest of the sample and is mostly found at the margins of the intercentrum, although more extensive remodeling and a greater abundance of secondary osteons can be clearly noted (Figs. 12E and 12F).

## Determination of ontogenetic stage

For this study, we utilized the formal histological ontogenetic stages (HOS) that were created for *Metoposaurus krasiejowensis* by *Konietzko-Meier, Bodzioch & Sander (2013)*. The organization of the periosteal bone is used to characterize the ontogenetic stage of an individual. HOS 1 lacks any periosteal ossification, HOS 2 features a wide (thick in transverse profile) periosteal cortex, HOS 3 features decreased vascularization in the external cortex, and HOS 4 features LAGs in the external cortex, specifically within the compact annulus (*Konietzko-Meier, Bodzioch & Sander, 2013*). Several other features also inform the relative maturity of the element, although they cannot be quantified in a fashion that would permit more discrete thresholds within the different HOS bins. Calcified cartilage is found in all of the samples, but it is more abundant in relatively immature specimens, particularly within the interior endochondral region. This is correlated with an increase in secondary remodeling of the trabeculae in this region. Additionally, the deposition of layers of lamellar bone within vascular canals increases in total thickness throughout ontogeny. Secondary osteons are rare throughout, but do occur more frequently in the most mature specimens.

Based on this classification system, we do not have any specimens referable to HOS 1 in the sample, as periosteal bone is clearly present in all intercentra. The majority of

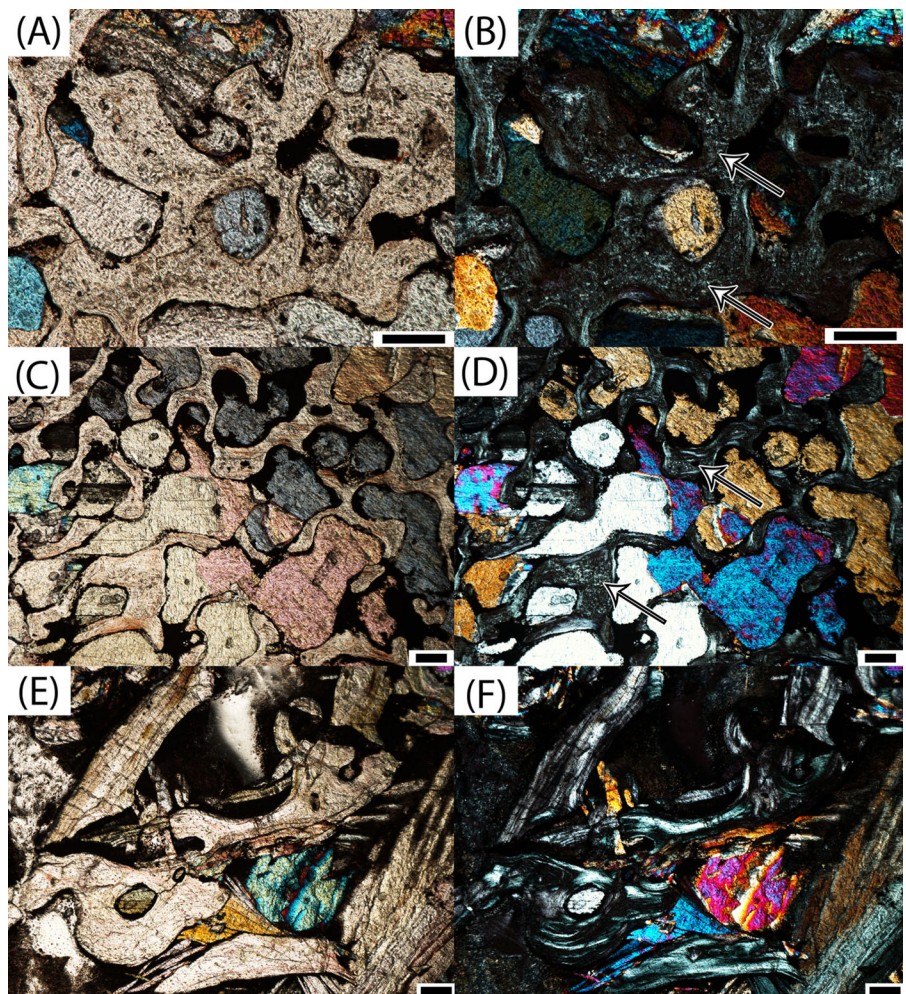

**Figure 12 Microphotographs of calcified cartilage and secondary remodeling in transverse profile.** (A) PEFO 36874a in the interior endochondral region; (B) calcified cartilage in the same specimen in polarized light; (C) PEFO 35392b in the interior endochondral region; (D) calcified cartilage in the same specimen in polarized light; (E) PEFO 38726 in the interior endochondral region; (F) absence of calcified cartilage and abundant secondary remodeling in the same specimen in polarized light. Arrows point to calcified cartilage scale bars = 100 μm.

specimens belong to HOS 2 (PEFO 36874a, PEFO 16696a, PEFO 4826, PEFO 16696b, PEFO 36874b); this designation is primarily made on the basis of relatively high vascularity within the external cortex. PEFO 38645 possesses a compact lamellar annulus, but no LAGs can be identified, rendering it referable to HOS 3 (Fig. 11A). Several of the other large specimens (PEFO 35392a, PEFO 16696c, PEFO 35392b) are referred to HOS 3 based on decreased vascularization of the external cortex, as well as the various changes in relative abundances of different tissues (Figs. 11B and 11C). We note that this classification is somewhat subjective because the distinction between HOS 2 and 3 (degree of vascularization of the cortex) is not absolute as in the distinction between HOS 1 and 2 (presence/absence of periosteal bone) or between HOS 3 and 4 (presence/absence of annulus with LAGs). These three specimens are more similar to each other than to any of the smaller intercentra, and they are more

**Table 2 Summary of major histological landmarks identified in the sampled specimens.**

| Specimen ID | Periosteal bone | Decreased vascularization in external cortex | Compact annulus | LAGs | HOS |
|---|---|---|---|---|---|
| PEFO 38726 | ● | ● | ● | ● | 4 |
| PEFO 4826 | ● | | | | 2 |
| PEFO 38645 | ● | ● | ● | | 3 |
| PEFO 36874a | ● | | | | 2 |
| PEFO 36874b | ● | | | | 2 |
| PEFO 35392a | ● | ● | | | 3 |
| PEFO 35392b | ● | ● | | | 3 |
| PEFO 16696a | ● | | | | 2 |
| PEFO 16696b | ● | | | | 2 |
| PEFO 16696c | ● | ● | | | 3 |

Notes:
For specimens with multiple elements, the listed order reflects their order by size, from smallest to largest.
Dots indicate the presence of the structure in specimens.

immature than PEFO 38645. PEFO 16696c is slightly more mature than either element of PEFO 35392 based on the degree of vascularization; the relative abundance of non-quantifiable features such as calcified cartilage, secondary remodeling, and lamellar deposition within the vascular canals is not appreciably different. Only the largest specimen, PEFO 38726, is referable to HOS 4, as it is the only specimen with a compact lamellar annulus in which LAGs can be tentatively identified (Fig. 11D). The ontogenetic assignments are summarized in Table 2.

# DISCUSSION

## Interpretation of ontogenetic maturity

The most significant finding of this study is the confirmation that, at least in the instance of the sample presented in this study, the small intercentra of proportions considered typical of *A. gregorii* are determined to belong to highly immature individuals. Several anatomical and histological features inform the ontogenetic assignment of these specimens: (1) a perforate notochordal channel; (2) a thick, more semi-circular periosteal region; (3) a greater abundance of calcified cartilage; (4) a minor degree of secondary remodeling; (5) relatively thin lamellar deposition within vascular canals; and (6) the absence of a compact lamellar annulus at the ventral margin (Figs. 8–12). All of these features are found in the three smallest intercentra (PEFO 36874a, PEFO 16696a, PEFO 4826) and provide insight into the ontogenetic changes in the internal structure of the axial column in metoposaurids.

We are confident that the open notochordal channel is a juvenile feature because its closure is widespread in Triassic temnospondyls, including metoposaurids (*Warren & Snell, 1991*). The notochordal channel is gradually reduced to a pair of perforations, one on each articular surface, that migrate dorsally and eventually disappear in some taxa (*Warren & Snell, 1991*; *Danto, Witzmann & Fröbisch, 2016*). Based on comparisons to described vertebral elements in *Metoposaurus krasiejowensis*, *Metoposaurus bakeri*,

*D. ouazzoui*, and isolated intercentra of *Koskinonodon perfectus*, this pattern often results in an entirely smooth articular surface with no notochordal perforation in mature individuals (*Case, 1932*; *Dutuit, 1976*; *Warren & Snell, 1991*; *Sulej, 2007*). Additionally, we can be certain that the notochordal channel does close in smaller individuals with elongate intercentra based on PEFO 36874a, which captures the onset of cartilaginous deposition into the canal and primary formation of endochondral trabeculae at the canal's margins; this is further discussed below (Figs. 8C and 8D). The designation of the three smallest intercentra as belonging to juvenile individuals is also supported by the presence of a highly vascularized periosteal region, which originates near the anteroventral and posteroventral margins, forming a shallower concave depression than the distinct triangle seen in larger intercentra of this study and the intercentra of *Metoposaurus krasiejowensis* (*Konietzko-Meier, Bodzioch & Sander, 2013*; Fig. 9). In these specimens, the apex of the periosteal region terminates well below the mid-height of the element owing to the notochordal canal (Figs. 7 and 9). Finally, the small intercentra show other evidence of a relatively immature ontogenetic stage, such as a high abundance of calcified cartilage, a high degree of vascularization in the external cortex, and minimal remodeling in the interior endochondral region (Figs. 11 and 12). As a result, we can be confident that the ossification of the notochordal channel did not occur relatively late in ontogeny and conclude that all three of the small intercentra belong to an early ontogenetic stage of a large metoposaurid rather than to a relatively late stage of the small *A. gregorii*.

Also of interest is PEFO 38645, which is characterized by a number of unusual features mentioned in the description. These include: (1) extreme thickening of the endochondral trabeculae at the articular surfaces; (2) the combination of a low periosteal apex with a compact lamellar annulus; (3) the absence of a clear periosteal cortex on the lateral margin in transverse profile; and (4) a thickened trabecular extension from the lateral margin to the inner endochondral region. This element was determined to be presacral, one of the posterior most elements sampled in this study, is of an intermediate size for the sample, and is of proportions typical of *Koskinonodon perfectus* (Table 1). It is notably larger than any intercentrum that has been ascribed to *A. gregorii* in the literature. One presacral intercentrum (UOPB 00118) of a much larger size was analyzed by *Konietzko-Meier, Bodzioch & Sander (2013)*, but it shows no evidence of sharing any of the unusual features we have identified in PEFO 38645 other than the possible extension of the periosteal cortex onto the dorsal margin of the element in transverse profile. This is also true when compared to the smaller PEFO 16696a, which was more tentatively identified as a presacral intercentrum. The combination of a periosteal apex that terminates well below the mid-height of the element and a compact lamellar annulus is also unlike any other intercentrum. Several hypotheses may explain these observations. One hypothesis is that the anomalous morphology is related to the serial position of this specimen within the presacral region. The marked thickening of the periosteal trabeculae on the articular surfaces and the apparent thinning of the periosteal cortex on the lateral margin may be related to biomechanical demands associated with intercentra in this region compared to the trunk region. The lack of similarity between this specimen and the presacral

intercentrum of *Konietzko-Meier, Bodzioch & Sander (2013)* may be related to the relatively high variability in external morphology within presacral vertebrae compared to other regions of the axial skeleton that is noted in *Metoposaurus krasiejowensis* and *D. ouazzoui* (*Sulej, 2007*; *Dutuit, 1976*). Given the different mechanical demands of each region, it seems probable that the developmental trajectories of different axial regions were at least slightly different with regards to the onset and degree of different processes. Timing of major events within the trajectory could also be influenced by intraspecific variation or developmental plasticity. Although the presence of a compact annulus suggests a more advanced ontogenetic stage, most of the features identified as informative for inferring the ontogenetic maturity suggest a relatively immature individual and are continuous with the gradual changes noted with increasing size in this sample. These include: (1) the absence of LAGs in the annulus (found in PEFO 38726); (2) a periosteal apex terminating well below the mid-height of the element; (3) a fairly high abundance of calcified cartilage; (4) a low degree of remodeling; (5) and thin lamellar deposition within the vascular canals. The overall vascularity of the periosteal cortex is difficult to evaluate considering that most of the tissue has either been destroyed postmortem or was otherwise reduced as the biological condition. It also cannot be ruled out that the specimen is in some way pathologic, although pathologies of extinct amphibians are extremely rare and have only been studied with regards to an anomalous external morphology (e.g., *Witzmann et al., 2013*). No abnormalities were apparent prior to the sampling of this specimen. Without additional sampling of this axial region, it is impossible to definitively conclude or exclude any of these proposed explanations. However, we do not consider this specimen to be contradictory to our general conclusions regarding the ontogenetic trajectory of metoposaurid intercentra given that most of the histological features are consistent with the trajectory of this sample.

The larger intercentra show little deviation from each other or from similarly sized and positioned intercentra that were sampled by *Konietzko-Meier, Bodzioch & Sander (2013)*, and any deviations are consistent with the ontogenetic trajectory seen in both samples. Unsurprisingly, both elements of PEFO 35392 are nearly identical in all regards, with the only differences being in the quality of preservation of the different trabecular regions. All of the large intercentra continue to show evidence of relative immaturity with regard to the same features as the smallest elements (calcified cartilage, vascularization, lamellar deposition with the canals, remodeling) up to the largest specimen, PEFO 38726 (Fig. 12). In this specimen, two continuous lines found in the compact annulus are tentatively interpreted to be LAGs (Fig. 11C). A significant gap in size exists between PEFO 38726 and the second largest element (PEFO 35392a), so it is likely that more gradual transitions were undetected in this sample. Although the material is from a variety of localities and stratigraphic horizons, we have found strong evidence that an increase in size of the intercentra results in progressively more advanced histological and anatomical features, leading us to conclude that that the sampled material can be compiled into a generalized composite growth series as it pertains to the development of the internal structures. There is no evidence for diminutive adults, as was proposed

to be the condition of *A. gregorii* (*Hunt, 1993*). However, we also note the limitations of our findings based on the isolated nature of the material. Although this is typical for North American metoposaurids, the axial position of each specimen could only be loosely identified, and it is highly unlikely that any two elements pertain to the same individual (other than in PEFO 35392). Furthermore, even intercentra of a nearly identical size, such as the three smallest specimens, may show some variation in anatomical and histological features due to intraspecific variation, developmental plasticity, or axial position, all of which serves to reinforce the importance of avoiding ontogenetic assignments based solely on absolute size. As a result, our study is unable to draw any robust conclusions regarding differences of the microanatomy and histology, either within an individual or throughout ontogeny, based on the axial position. We did not sample cervical, atlas-axis, or caudal intercentra, and the majority of those sampled are from the anterior to mid-trunk region. Any interpretations of variation between samples of a comparable size with regard to features that can only be relatively compared (e.g., abundance of calcified cartilage) would require higher sample sizes and narrower constraints on the axial position than is possible from a sample of completely isolated elements.

Finally, one feature that does not appear to be ontogenetically informative in this sample, but that is nonetheless noteworthy is the open cavities found throughout the periosteal cortex in many of the intercentra (Fig. 13). These cavities were only figured in the presacral intercentra of *Metoposaurus krasiejowensis* (*Konietzko-Meier, Bodzioch & Sander, 2013*), but in this sample, there is no apparent pattern in the distribution or relative abundance based on axial position. Because of their small size and infilling with secondary minerals in many specimens, they are probably impossible to consistently detect when examining the external morphology of the intercentrum. The cavities likely represent nutrient foramina whose trajectories lie in more than one orthogonal plane, hence why some cavities appear to terminate shallowly in the periosteal cortex while others penetrate to the inner endochondral region, as well as why some of the foramina appear L-shaped in sagittal profile rather than simple straight perforations. Without a high-resolution three-dimensional analysis of the intercentra, it is impossible to discern if there is any pattern in the trajectory, distribution, or morphology of these foramina within intercentra, across the axial skeleton, or throughout ontogeny. No such patterns appear to exist in this sample. The foramina appear to remain of a constant size throughout the growth of the intercentrum, as they are proportionately smaller in larger specimens.

## Ontogenetic trajectory of large North American metoposaurids

Our findings are consistent with the hypothesis that all intercentra of this sample pertain to various ontogenetic stages of a large metoposaurid. Because *K. bakeri* has not been identified west of Texas, and its intercentra differ from those of *Koskinonodon perfectus* with regard to the notochordal channel (discussed below), we tentatively assign the PEFO material to *Koskinonodon perfectus*, with the understanding that future revision may be necessary as more diagnostic material is recovered (*Hunt, 1993*;

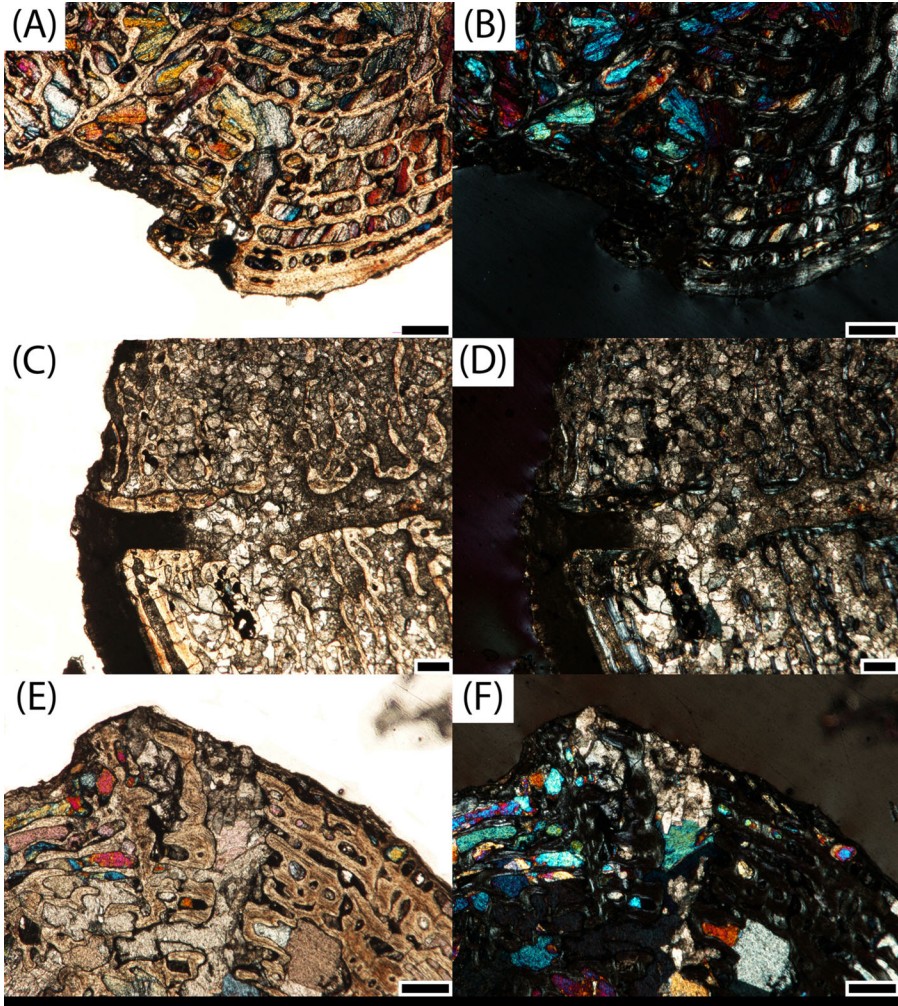

**Figure 13 Microphotographs of nutrient foramina in transverse profile.** (A) Nutrient foramen from the ventral periosteal surface PEFO 4826; (B) same specimen in polarized light; (C) nutrient foramen from the lateral surface of PEFO 16696c; (D) same specimen in polarized light; (E) nutrient foramen from the dorsolateral surface of PEFO 36874a; (F) same specimen in polarized light. Scale bars = 500 μm.

*Long & Murry, 1995*). It cannot be excluded that material from the Petrified Forest Member could belong to a different taxon of large metoposaurid that is presently unknown as the result of a paucity of such individuals in the upper units of the Chinle Formation. It is possible that the onset of ossification of the notochordal channel reflects a milestone in the development of *Koskinonodon perfectus*. In light of the hypothesis suggesting that *Koskinonodon* could have had ecologically separated life stages (*Rinehart, Lucas & Heckert, 2009*) and arguments by previous workers that the morphology of *A. gregorii* is more suited for a terrestrial lifestyle (*Hunt, 1993*; *Spielmann & Lucas, 2012*), the ossification of the notochordal canal could potentially represent the onset of a fully aquatic lifestyle if small metoposaurids were semi-aquatic. This study has also produced an unexpected finding that suggests some differences in timing of the ontogenetic trajectory of *Koskinonodon perfectus* in relation to other

metoposaurids with known vertebral columns. In the original description of *K. bakeri*, *Case (1932)* noted that the presence of a notochordal channel and its persistence as reduced perforations on the articular surfaces in more mature specimens differed from other metoposaurid specimens from Texas, presumably of *Koskinonodon perfectus*, in that the known material of the latter lacked any sort of perforation. The absence of any perforations appears in the intercentra of *Koskinonodon perfectus* that are described or figured in other publications (e.g., *Colbert & Imbrie, 1956*; *Hunt, 1993*; *Long & Murry, 1995*; *Spielmann & Lucas, 2012*), and we have also found this same absence in all of the intercentra sampled here. These observations suggest that with regards to timing, the ossification of the notochordal canal occurs much earlier in *Koskinonodon perfectus*. We note that the smallest specimen analyzed by *Konietzko-Meier, Bodzioch & Sander (2013)*, an early juvenile (UOPB 00117), is larger than two of the three small intercentra sampled here (PEFO 16696a, PEFO 36874a) but is classified as being more ontogenetically immature (HOS 1) than either specimen due to the absence of periosteal ossification (Figs. 9A and 9B; Table 2).

PEFO 36874a is of particular interest in this regard as it is clearly in the process of undergoing ossification of the notochordal channel, even though it is the smallest intercentrum in this sample (Figs. 8C and 8D). This was not evident when examining the external morphology of the specimen, as the notochordal channel or pit is usually filled with secondary minerals. Precursory cartilage can be clearly seen growing into the channel near the geometric center, with some endochondral trabeculae forming in the wake of this deposition (Fig. 4B). The dorsal half appears to be contributing more material through bone deposition, but this requires additional specimens to verify (Fig. 4B). Although this specimen is the largest of the three intercentra with notochordal canals, this does not contradict our ontogenetic assignment based on an examination of the external morphology of other small, elongate intercentra at PEFO. There appears to be some variability in the exact timing of the closure of the notochordal channel, as specimens of similar size and proportion exhibit the full range of conditions, from an open channel to a smooth articular surface lacking any trace of the channel. This could be owing to a number of processes that require additional samples to evaluate, such as the progression of ossification of the vertebral column in the anterior-posterior direction or intraspecific variation in the onset of ossification. If the early stages of vertebral ossification were in some way influenced by environmental factors rather than the absolute size of the animal, developmental plasticity, which occurs in both extant and extinct amphibians, could explain how intercentra could sometimes be histologically more immature than slightly smaller ones (*Newman, 1992*; *Schoch, 2014*). As previously noted, this may also indicate a relatively fast ossification of the notochordal channel.

It may be that *Koskinonodon perfectus* juveniles experienced a relatively rapid burst of early endochondral ossification within the skeleton in comparison to *Metoposaurus krasiejowensis*, such that even a slight difference in absolute size between two elements could be characterized by markedly different internal anatomy, such as the differences in notochordal canals of the four smallest specimens (Figs. 8 and 9). As previously

discussed in other sections, this could reasonably result from intraspecific variation, developmental plasticity, or axial position.

Finally, only the largest intercentrum sampled in our study (PEFO 38726) contains possible LAGs in the external cortex (Fig. 11D). This element is most comparable in size to UOPB 00115, which was classified as a late juvenile of HOS 3 (*Konietzko-Meier, Bodzioch & Sander, 2013*) and in which no LAGs were observed. This suggests that *Koskinonodon perfectus* may have reached maturity at a smaller size than *Metoposaurus krasiejowensis*, but again, additional sampling is required. Variability in ontogenetic trajectories has been previously documented between *D. ouazzoui* and *Metoposaurus krasiejowensis* and was proposed to be the result of differing environmental conditions (*Konietzko-Meier & Klein, 2013*). As the Chinle depositional basin was positioned closer to the equator in comparison to the environments in which *D. ouazzoui* and *Metoposaurus krasiejowensis* are found (e.g., *Atchley et al., 2013*; *Steiner & Lucas, 2000*; *Rowe et al., 2007*; *Zeigler & Geissman, 2011*; *Nordt, Atchley & Dworkin, 2015*), it is plausible that the paleoenvironment differed sufficiently from that of both taxa so as to result in a distinct ontogenetic trajectory in *Koskinonodon perfectus*. Additional sampling of material, particularly limb elements, is needed for comparative analyses to assess this possibility.

## Implications for metoposaurid paleobiology

These findings, particularly with regard to the smallest intercentra, provide one line of evidence to support a hypothesis of niche partitioning between life stages of metoposaurids, which has been suggested in *Koskinonodon* (*Rinehart, Lucas & Heckert, 2009*) and in *Metoposaurus* (*Sulej, 2007*). Such partitioning could reasonably have created an associated taphonomic bias, which is well documented in both dense bonebeds and localities with more dispersed material. All known metoposaurid bonebeds have so far produced only large, relatively mature individuals with no evidence of the earliest ontogenetic stages (*Case, 1932*; *Colbert & Imbrie, 1956*; *Dutuit, 1976*; *Hunt, 1993*; *Sulej, 2007*; *Lucas et al., 2010*; *Brusatte et al., 2015*). Furthermore, although material from mature individuals of *Koskinonodon perfectus* is common in the middle Norian, material referable to juveniles of the taxon is extremely rare, providing another line of support for niche partitioning; to date, only two partial skulls have been described in detail (*Zanno et al., 2002*; *Gee & Parker, in press*). A juvenile specimen of *Metoposaurus bakeri* from Nova Scotia was first noted by *Baird (1986)* but has only been briefly described by subsequent authors (*Hunt, 1993*; *Sues & Olsen, 2015*). Material of *A. gregorii* is common in the Redonda Formation in New Mexico but occurs mostly within a single quarry (Gregory Quarry, NMMNH locality 485) (*Spielmann & Lucas, 2012*). As a result, the relative abundance of *A. gregorii* may not be the result of ecological turnover as postulated by *Hunt (1993)*, but may represent the preservation of depositional environments inhabited by juveniles of *Koskinonodon perfectus*. As bonebeds of mature metoposaurids have been interpreted as evidence of ecological aggregation prior to death, it is not implausible to infer that juveniles may also have naturally aggregated separately from adults, creating a preservation potential for dense assemblages of one growth stage (*Lucas et al., 2010*; *Brusatte et al., 2015*). Based on the isolated and disarticulated nature of most *Apachesaurus*

material, we do not believe these deposits represent mass mortality events, but that they are more likely representative of depositional environments frequented by small metoposaurids over longer durations of time. This hypothesis is supported by a previous study that surveyed blue paleosol localities at PEFO and found that material of many rare taxa, as well as that of *A. gregorii*, are found mostly within these uncommon horizons (*Loughney, Fastovsky & Parker, 2011*). PFV 040, PFV 215, and potentially PFV 122, the three localities from which specimens for this study were sourced, are all blue paleosol horizons. This lithology is interpreted to have formed in low-energy systems, primarily abandoned channels and ponds adjacent to the main river channel, in contrast to the dominant red floodplain deposits in which fossil material is more fragmentary and isolated (*Loughney, Fastovsky & Parker, 2011*). The association of *Apachesaurus* material with these blue paleosol localities supports the hypothesis that deposits that are disproportionately skewed toward fossils of small-bodied metoposaurids (exemplified by PFV 040 and PFV 215) and potentially form in different geologic settings than deposits that are skewed toward large metoposaurids. This in turn supports the hypothesis of natural ecological separation between life stages of metoposaurids. Additionally, taxa that are primarily associated with blue paleosol horizons may not be as stratigraphically restricted as previously thought, and a perceived faunal turnover may in fact be more closely linked to changes in the relative likelihood of preservation of different depositional settings.

## CONCLUSION

These findings reiterate the importance of evaluating the potential for morphological variation to be the result of ontogeny, especially when comparing two taxa of vastly different sizes, such as *A. gregorii* and *Koskinonodon perfectus*. Although fossils of *A. gregorii* are common in late Norian deposits, the vast majority of this material has consisted of isolated elongate intercentra, which we demonstrate here cannot be considered apomorphic. Limited fragmentary pectoral and pelvic material of *A. gregorii* has been described in the literature, but no justification for ascribing it to the taxon has ever been provided (e.g., *Hunt, 1993*; *Long & Murry, 1995*; *Spielmann & Lucas, 2012*). Although this material was recovered from the same quarry as cranial and vertebral material, there is no published work suggesting that any of it was found in articulation with any of the diagnostic cranial material (*Spielmann & Lucas, 2012*). North American metoposaurid specimens are frequently isolated or disarticulated, but this does not negate the importance of reevaluating the taxonomic identity of this material to determine whether they preserve robust diagnostic traits. It is possible that these assignments were made solely on the basis of diminutive size (e.g., *Hunt, 1993*; *Long & Murry, 1995*; *Spielmann & Lucas, 2012*), which cannot be utilized as in species differentiation given the role of ontogeny in producing morphological variation associated with different size bins (*Steyer, 2000*; *Horner & Goodwin, 2009*; *Witzmann, Scholz & Ruta, 2009*). Similarly, although a large number of diagnostic cranial characters have been identified for *A. gregorii*, only a single character, the shallow otic notch, can be confirmed in any specimens beyond the holotype (*Spielmann & Lucas, 2012*). The potential for these

cranial landmarks to be ontogenetically influenced has not been sufficiently addressed by past workers, in spite of the widespread documentation of morphological changes associated with ontogeny in both extant and extinct amphibians (*Hanken, 1992*; *Fröbisch et al., 2010*; *Schoch, 2014*). For example, studies of other Triassic temnospondyls have shown that the otic notch, occipital condyles, and cultriform process (by virtue of its relationship with the interpterygoid vacuities) all play a role in bite force mechanics (*Fortuny et al., 2012*, *2016*; *Lautenschlager, Witzmann & Werneburg, 2016*). Based on these findings, the presence of shallow otic notches, reduced projection of the occipital condyles, and a wider cultriform process (all supposedly diagnostic traits of *A. gregorii*) may in fact be influenced by changing biomechanical demands throughout ontogeny, rather than being the result of speciation. The potential for intraspecific variation to exert an influence on metoposaurid morphology has also not been well studied in North American taxa even though studies of bonebeds of *Metoposaurus krasiejowensis* and *Metoposaurus algarvensis* have demonstrated a higher degree of variability in many cranial regions than previously thought (*Sulej, 2007*; *Brusatte et al., 2015*).

Finally, we believe that our results provide one line of evidence that *A. gregorii* is not in fact a distinct species, but rather that it is an early ontogenetic stage of a large metoposaurid, such as *Koskinonodon perfectus*. The stratigraphic distribution that is alleged to reflect ecological turnover could actually be controlled by taphonomic bias if niche partitioning characterized different life stages of *Koskinonodon perfectus*. The role of ontogeny and intraspecific variation in producing morphological variation in features such as cranial suture patterns, the basicranium, and the otic notch remain relatively unexplored in North American metoposaurids. Discovery and study of additional juvenile specimens is needed to establish a more robust ontogenetic characterization of the earliest stages of metoposaurid development, but our study has also demonstrated that underutilized methods of analysis such as paleohistology on existing specimens can shed new light on the paleobiology of extinct taxa with implications for taxonomy and ontogeny.

## INSTITUTIONAL ABBREVIATIONS

**NMMNH**    New Mexico Museum of Natural History and Science, Albuquerque, NM, USA

**PEFO**    Petrified Forest National Park, AZ, USA

**UOPB**    University of Opole, Department of Biosystematics, Opole, Poland

## ACKNOWLEDGEMENTS

Thanks to Matthew Smith (PEFO museum curator) for providing access to specimens for histological analysis and to Brad Traver (PEFO superintendent) for granting permission to conduct the destructive analyses. Thanks to Cathy Lash (PEFO fossil preparator) for assistance with molding and casting of the specimens and to Yara Haridy (University of Toronto) for instruction and assistance with the preparation and imaging of thin sections. Thanks to the reviewers, Dorota Konietzko-Meier, Alexandra

Houssaye, and Tomasz Sulej, and to the editor, Andrew Farke, for providing insightful comments that greatly improved this manuscript. Any opinions, findings, or conclusions of this study represent the views of the authors and not those of the U.S. Federal Government. This is PEFO Paleontological Contribution no. 49.

### Funding

This work was supported by a summer internship grant by the Petrified Forest Museum Association (PFMA) to BMG. The funders had no role in study design, data collection and analysis, decision to publish, or preparation of the manuscript.

### Grant Disclosures

The following grant information was disclosed by the authors:
Petrified Forest Museum Association (PFMA).

### Competing Interests

The authors declare that they have no competing interests.

### Author Contributions

- Bryan M. Gee conceived and designed the experiments, performed the experiments, analyzed the data, wrote the paper, prepared figures and/or tables, reviewed drafts of the paper.
- William G. Parker contributed reagents/materials/analysis tools, wrote the paper, prepared figures and/or tables, reviewed drafts of the paper.
- Adam D. Marsh conceived and designed the experiments, contributed reagents/ materials/analysis tools, wrote the paper, reviewed drafts of the paper.

### Data Availability

Data supplied for general background purposes is present in the manuscript in Tables 1 and 2.

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
