# Peer review of "Microanatomy and paleohistology of the intercentra of North American metoposaurids from the Upper Triassic of Petrified Forest National Park (Arizona, USA) with implications for the taxonomy and ontogeny of the group"

_PeerJ, doi:10.7717/peerj.3183_

## Round 0.1 · original submission · Minor Revisions

· Academic Editor

Minor Revisions

The reviewers provide a number of suggestions for revision of the paper, particularly in some areas to provide additional data (e.g., on the histology) and discussion. Please incorporate these into the paper to the best of your ability (with the exception noted below).

Other items of note:

- I would recommend leaving the methods and results sections at their current length rather than trimming them (contra two of the reviewers' suggestions). This level of detail is appropriate for the paper, in my opinion, and it is useful to include the specific methods with the paper to avoid having to search across the literature.

- Two of the reviewers recommend including some additional context in the introduction; this should be included if possible.

·

Basic reporting

Dear Editor in Chief of PeerJ,
I have reviewed the manuscript entitled “Paleohistology of the intercentra of North American metoposaurids from the Upper Triassic of Petrified Forest National Park (Arizona, USA) with implications for the taxonomy and ontogeny of the group” by Bryan M. Gee and colleagues.
It is a very interesting study about bone histology of the two North American metoposaurids. It is especially important, because the histology of the American taxa is absolutely unknown up to date. The authors based on the bone histology of intercentra tried to solve the taxonomical problem with the small taxon Apachesaurus. According to some authors Apachesaurus is only a juvenile form of one of the larger taxa, according to others it is a separate taxon. Moreover the authors analyzed rate of ossification processes and provided some conclusion about the development and biology of life.
As the histology is a destructive method and access to postcranial elements of Temnospondyli is highly limited each contribution is really welcomed. Although the issue is really interesting and deserves to be published in a high ranked journal as PeerJ, I found some problems that must be assessed before final acceptation of the paper (for details see manuscript).

1. Introduction:
For not-temnospondyl specialists it might be good to introduce short metoposaurids and American taxa with the specification of the problem about the position of Apachesaurs, including both hypothesis and arguments pro and against (juvenile Koskinodon vs adult dwarf). Now it is not clear, if you are speaking about problems with diagnose for this particular locality or if it is a general problem with the A. gregorii. In my opinion the short chapter about the diagnostic characters (localities, most important skull characters) of both taxa might help to understand the problem with the ontogenetic or/and taxonomic variety. Also you may list, what kind of other bones (diagnostic or not) suspected to be an Apachesaurus, you have got in Petrified Forest National Park (not only to give the citations – again for not temnospondyl researcher will be easier to have all information in one place and not to search for specific sources).
This chapter you should change a little bit to add some more citations about the history of the histological temnospondyl studies, especially based on long bones. Now you have cited only three papers about long bones, but there is no clear why have you selected these three as examples (all are interesting, but not more important than others). Try to rewrite this paragraph to make it more general.
No clear goal/-s of the study.
2. Material and methods:
Maybe you can make the description shorter, because the methodology of the producing thin-sections is quite well known and described in the methodological publications. But you should add some details about the HOS determination.
3. Results:
It is always helpful for readers to have the short morphological description. Maybe you can add the short chapter with the measuring results (which you have now in method description) and the most important morphological characters. I still think that it is important to add here the preliminary taxonomical diagnosis (why do you think that this specimen belongs to… I know that only one belongs to A.g., but proof please why) – then you can discuss the histology with the morphological signal.
This is mostly description of microanatomy, not of the histology. But in my opinion the histological description (the organization of the collagen fibers, resorption process, primary or/and secondary osteons/tissue, remains of calcified cartilage, Sharpeys Fibers, etc. are very important for the deduction about the ontogenetic stage).
I would suggest also the other organization of the description of specimens. Maybe you should add here the preliminary taxonomical determination of specimens based on the morphology. And then describe the specimens in two groups, in each from the smallest. I also do not like the “group” specimen numbering. It is a little surprising that you have two or three bones, very different about size (so two/three specimens) with the same number (i.e. Fig 3D- E and Fig. 3H-J). Of course collection numbers are independent and you cannot change these, but maybe you can add to each bone with the same number a letter: A,B,C and then describe each bone separately (as a separate specimen - not as a group).
4. Discussion and conclusions:
I am not convinced. In my opinion you need the classical histological analysis, not only microanatomy. Maybe one of this smallest has ontogenetically old tissue, strongly secondary. How does it look with the remains of calcified cartilage in the “endochondral” domain? Stereospondyli are known for the log preservation of c.c., but the amount of cc. decrease during ontogeny (see the paper about Metoposaurus and Konietzko-Meier et al., 2014). It is not excluded that the one Apachesaurus-like intercentrum has similar microstructure, but on the histo-level is different. If the Apachesaurus is a separate taxon, dwarf-metoposaurus, the growth should terminate earlier, with the same growth pattern (but then the tissue is older) or be slower. There are a lot of papers about histology of dwarf-amniots.
My first take based on the title and introduction was that you want to use only histology to confirm (or not) the taxonomical assignment of a long intercentrum and try to estimate some ontogenetic processes. However, the most important histological part seems to be the weakest from the whole chapter (see comments). To proof your taxonomical diagnose you should add more histological details. Maybe on this level the intercentra look different. And for Temnospondylli especially important seems to be the analysis of the structure in polarized light. Some details are visible only in this light and only then you can see “the second face” of histological structures. But in generally in discussion you have a lot of interesting conclusions and hypothesis. Thus you should add some information to the introduction and precise few new goals.
The English of the present version is correct.

Experimental design

Standard procedures are used to assess the bone histology. The images provided should be greatly improved. The authors should add more labelling. See annotations on the manuscript.

Validity of the findings

Some of the conclusions should be better discussed. Also in my opinion the classical histological (not microstructural) analysis may bring a lot of important information. The sections are ready (and the bones are destroyed) and should be “used” as much as possible. See also Basic report.

Additional comments

I have made all comments directly on the manuscript.

·

Basic reporting

no comment

Experimental design

no comment

Validity of the findings

no comment

Additional comments

Dear Editor/Authors

The question raised by this manuscript is interesting since, indeed, possible ontogenetic variation is sometimes not taken into consideration with the erection of a new species.
However the paper would benefit from some rewriting. Indeed more details about the data available to distinguish the various metoposaurid taxa would be appreciated in the introduction in order to better define the context/state of the art of the question.
Conversely, the method & result sections are too long and should be shortened. Moreover the description is not always clear with appropriate terms and illustrations are sometimes not sufficient for what is claimed in the text.
I put several comments directly in the pdf. Major concerns deal with intracolumnar variability which is never evoked. Some hypotheses could also appear less as claims since it is based on relatively limited data. But the discussion is interesting. It will also be great to check the text to clearly state what could be size related and what could really be ontogenetic.
I thus recommend a moderate revision.
Best wishes,

Alexandra Houssaye

·

Basic reporting

No comment

Experimental design

No comment.

Validity of the findings

No comment

Additional comments

Congratulations. It is important paper for all who works on temnospondyls. Almost all is very good, I have only some small comments to the text in the attached PDF.

---

## Round 0.2 · Minor Revisions

· Academic Editor

Minor Revisions

Thank you for your thorough response and revision in light of the reviewers' comments. I have only a few relatively minor suggestions (all stylistic and cosmetic) that I would like to see incorporated before my final decision. These are outlined below:

- The first paragraph of the introduction is _really_ long, and needs to be split up into two or three for easier reading. You might also consider splitting the first paragraph in materials & methods and the discussion, too.
- line 63: "intended to be addressed in future work" -- I would recommend removing this phrase; it is sufficient just to end the sentence with the statement that an analysis of "M." bakeri is outside the scope of the current paper.
- line 75: Change "while" to "Although" -- "while" has a time connotation.
- line 90: the studies cited here excellent but are all in archosaurs; is there a study or two on other tetrapods that could be referenced?
- line 203, 204: Please include the years for the NPS citations here.

---

## Round 0.3 · accepted · Accept

· Academic Editor

Accept

Thank you for your thorough attention to the previous round of comments. In my estimation, your paper is ready for publication.